# Self-adaptive amorphous CoOₓClᵧ electro-catalyst for sustainable chlorine evolution in acidic brine

Mengjun Xiao [1,10], Qianbao Wu [1,10], Ruiqi Ku[2], Liujiang Zhou [3], Chang Long [1], Junwu Liang [4,5] ✉, Andraž Mavrič [6], Lei Li [1], Jing Zhu[7], Matjaz Valant [6], Jiong Li[8], Zhenhua Zeng [9] & Chunhua Cui [1] ✉

Electrochemical chlorine evolution reaction is of central importance in the chlor-alkali industry, but the chlorine evolution anode is largely limited by water oxidation side reaction and corrosion-induced performance decay in strong acids. Here we present an amorphous CoOₓClᵧ catalyst that has been deposited in situ in an acidic saline electrolyte containing $Co^{2+}$ and $Cl^-$ ions to adapt to the given electrochemical condition and exhibits ~100% chlorine evolution selectivity with an overpotential of ~0.1 V at 10 mA cm$^{-2}$ and high stability over 500 h. In situ spectroscopic studies and theoretical calculations reveal that the electrochemical introduction of $Cl^-$ prevents the Co sites from charging to a higher oxidation state thus suppressing the O-O bond formation for oxygen evolution. Consequently, the chlorine evolution selectivity has been enhanced on the Cl-constrained Co-O* sites via the Volmer-Heyrovsky pathway. This study provides fundamental insights into how the reactant $Cl^-$ itself can work as a promoter toward enhancing chlorine evolution in acidic brine.

Chlorine ($Cl_2$) gas is an essential chemical with an annual production of over 70 million tons for a wide range of applications in water treatment, organic synthesis, and polyvinyl chloride production[1–3]. It is mainly produced through electrochemical chlorine evolution reaction (CER)[4] in the chlor-alkali industry[5–7], where the oxygen evolution reaction (OER) can compete with CER in aqueous electrolytes owing to their comparable onset potentials and similar types of catalytic active sites[8–10], thereby it is a great challenge to ensure the highly selective production of $Cl_2$ gas. The state-of-the-art industrial electrolyzers for CER generally use relatively acid-resistant mixed metal oxides based on high-cost precious metal (Ru or Ir)[11,12], which are highly active for OER as well. To ensure the high efficiency in the CER system and improve the $Cl_2$ selectivity, a high molar concentration of $Cl^-$ ion (5.0 M) and acidic pH conditions are required[13–15]. However, for practical applications, Ru-based CER catalysts are still insufficiently stable under long-term operation due to the formation of soluble ruthenium chloride[16]. Hence, some additives or dopants, such as the SnOₓ, were used to improve the activity as well as stability of mixed metal oxides-based CER catalysts over the past decades[17–19].

[1]Molecular Electrochemistry Laboratory, Institute of Fundamental and Frontier Sciences, University of Electronic Science and Technology of China, Chengdu 610054, China. [2]School of Physics, Harbin Institute of Technology, Harbin 150001, China. [3]School of Physics, University Electronic Science and Technology of China, Chengdu 611731, China. [4]Optoelectronic Information Research Center, School of Physics and Telecommunication Engineering, Yulin Normal University, Yulin, Guangxi 537000, China. [5]Center for Applied Mathematics of Guangxi, Yulin Normal University, Yulin, Guangxi 537000, China. [6]Materials Research Laboratory, University of Nova Gorica, Vipavska 13, SI-5000 Nova Gorica, Slovenia. [7]Department of Chemical Physics, School of Chemistry and Materials Science, University of Science and Technology of China, Hefei 230026, China. [8]Shanghai Synchrotron Radiation Facility, Shanghai Advanced Research Institute, Chinese Academy of Sciences, Shanghai 201210, P. R. China. [9]Davidson School of Chemical Engineering, Purdue University, West Lafayette, Indiana 47907, USA. [10]These authors contributed equally: Mengjun Xiao, Qianbao Wu. ✉e-mail: jwliang@ylu.edu.cn; chunhua.cui@uestc.edu.cn

The CER occurs under harsh reaction conditions according to the Pourbaix diagram of an aqueous saline electrolyte[20–22], especially at pH <3 with a highly polarized potential range, unavoidably leading to the dissolution of most metal oxide materials. Recently, low-cost non-precious metal oxides, such as highly crystalline $Co_3O_4$ and transition-metal antimonates[23–26], have been tested as CER catalysts, which display decent stability since their dense, thick, and crystalline natures somehow prevent their complete dissolution in corrosive and acidic electrolytes. By contrast, an amorphous nonprecious metal oxide film can dissolve in such electrolytes in tens of minutes[27]. As a result, the intrinsic electrochemical surface amorphization upon anodic polarization is very detrimental to the catalytic stability but technically inevitable. Despite the beneficial effect of enriching the surface-active sites through amorphization, it remains a huge challenge to exploit amorphous non-precious metal oxide catalysts, like the alkaline OER catalysts, toward highly active, selective, and sustainable CER.

In this work, we present a self-adaptive amorphous $CoO_xCl_y$ catalyst, which was in situ deposited on a bare F-doped tin oxide (FTO) electrode at pH ≤ 2 in saline solutions containing $Co^{2+}$ and $Cl^-$ ions. The final state of the deposited film catalyst depends on a given electrochemical condition, including the applied potential, solution pH, and electrolyte concentration[28]. We show that CER takes place heterogeneously on the in situ deposited $CoO_xCl_y$ catalyst. We find that an anodic polarization at ≥ 1.67 V (versus RHE if not otherwise noted) leads to a moderate deposition rate of $CoO_xCl_y$ and the $Cl^-$ in electrolytes aids the co-deposition of Co and Cl into the catalyst film. Experimental results and density functional theory (DFT) calculations show that the introduction of $Cl^-$ can share the oxidative equivalent under highly anodic potential and inhibit the oxidation of Co to a higher level for OER, leading to increased selectivity for CER. Online differential electrochemical mass spectrometry (DEMS) and rotating ring-disk electrode (RRDE) verified the efficient CER with ~100% selectivity in 0.5 M $Cl^-$ brine solution. A certain time of catalyst deposition at 1.67 V leads to the CER activity increasing to around 10 mA cm$^{-2}$ and afterwards a slight increase over 500 h.

## Results and Discussion
### Screening out metal cations for CER in acidic brine
We first screened several transition metal cations, such as nickel (Ni), manganese (Mn), iron (Fe), copper (Cu), and cobalt (Co) at pH 2 for CER. We demonstrated that adding $Cl^-$ ions into Mn, Ni, and Cu-containing brine solutions did not obviously shift the onset potential relative to the $Cl^-$-free electrolytes with the same ionic activity. Interestingly, a substantial decrease in the onset potential and an evident increase in current density were observed in the saline solution containing 0.001 M $Co^{2+}$ or $Fe^{3+}$ and 0.5 M $Cl^-$ relative to that containing 0.001 M $Co^{2+}$ or $Fe^{3+}$ and 0.5 M $ClO_4^-$ (Supplementary Fig. 1). However, although the Fe-catalyst displayed initial CER activity, the subsequent electrodeposition was not favorable, regardless of either increasing $Fe^{3+}$ or $Cl^-$ concentration thus being precluded (Supplementary Note 1 and Supplementary Figs. 2–6). In contrast to the large onset difference between CER and OER over deposited Co-catalyst, the dimensionally stable anode (DSA) and $RuO_2$ catalysts loaded on FTO showed close onset potentials in 0.5 M NaCl electrolyte for CER relative to the $Cl^-$-free electrolyte for OER (Supplementary Fig. 7). Meanwhile, we precluded the anion influence from Co precursors on cyclic voltammograms (Supplementary Fig. 8). In addition, we found that the current densities increase with the concentration of either $Co^{2+}$ or $Cl^-$ (Supplementary Fig. 9). The X-ray fluorescence (XRF) Co Kα mapping showed that the amount of deposited Co sites increases with the $Cl^-$ concentration, suggesting the interplay between $Co^{2+}$ and $Cl^-$ (Supplementary Fig. 10).

The linear scan voltammetry (LSV) curve after 2 h of electrodeposition at 1.67 V in $Cl^-$-containing electrolyte was used as an example to exhibit a substantial increase in current density and a downshift of onset potential over 400 mV at 10 mA cm$^{-2}$ relative to that of $Cl^-$-free electrolyte (Fig. 1a and Supplementary Fig. 11). The in situ deposition of $CoO_xCl_y$ at 1.67 V was accompanied by $Cl_2$ production (Supplementary Fig. 12), as well as reduced charge transfer resistance (Supplementary Fig. 13). As expected, a stepwise increase in potentiostatic time brought in an increased mass loading of the Co sites, which was verified by the time-dependent Co Kα XRF test (Supplementary Fig. 14). Accordingly, the continuous increase of active Co sites led to a gradual attenuation of overpotential from -0.4 V after the first potential scan to finally -0.1 V (Fig. 1b). During this process, the catalyst film presents an epitaxial-growth-like behavior, of which the applied potential at the electrode/electrolyte interface can oxidize the $Co^{2+}$ for the deposition of outermost catalyst layer until the interfacial potential drop across the catalyst[29,30] cannot support the subsequent oxidation-deposition of $Co^{2+}$.

### Electrochemical CER selectivity and stability
Achieving high selectivity for CER is crucial due to the competing OER[31–33]. Thus, we evaluated the CER selectivity based on the rotating ring-disk electrode (RRDE), where the generated $Cl_2$ on the disk catalyst electrode was quantified through electrochemical reduction on the ring electrode (Supplementary Fig. 15 and Supplementary Note 2). Through comparing the onset potentials for both $O_2$ and $Cl_2$ reduction on the Pt ring electrode based on the established method[34], 0.95 V has been applied for $Cl_2$ reduction and quantification (Supplementary Fig. 16 and Supplementary Note 3). The CER selectivity was evaluated by the chronoamperometry method at indicated potentials using RRDE[35] that was conducted in an Ar-saturated electrolyte containing 0.1 M $Co^{2+}$ with various concentrations of $Cl^-$ at pH 2 at 1600 RPM (Supplementary Figs. 17–21). Significantly, in acidic saline containing 0.1 M $Co^{2+}$ and 0.5 M $Cl^-$, a CER selectivity of ~100% at 1.67 V was achieved (Fig. 1c). It is worth noting that the $Cl_2$ selectivity already reached 90 ± 1.5% at 1.67 V when the $Cl^-$ concentration is as low as 0.01 M. Importantly, in 0.5 M $Cl^-$, the CER selectivity can easily achieve 100% at a wide range of potentials. These results are in line with the substantially reduced onset potential for CER in comparison to that for OER (Fig. 1a). Compared to the $CoO_xCl_y$ catalyst, the selectivity of DSA (~80%) and $RuO_2$ (~88%) catalysts were distinctly low even in 0.5 NaCl electrolyte (Supplementary Fig. 22).

To further prove the CER selectivity, online DEMS was applied to discriminate the CER and OER[36–38]. The DMES experiment was implemented in a three-electrode H-cell in acidic saline solutions containing 0.1 M $Co^{2+}$ and 0.5 M $Cl^-$ (Supplementary Fig. 23). Significantly, as shown in Fig. 1d, the DEMS data clearly exhibit that the $^{70}Cl_2$ gas was the only product in 0.5 M $Cl^-$ electrolyte during the CER process, and no other signals were detected, such as $^1H^{16}O^{35}Cl$, $^{35}Cl^{32}O_2$ or $^{35}Cl^{16}O$ (Supplementary Fig. 24). In contrast, in the absence of $Cl^-$, the $^{32}O_2$ was the exclusive product (Supplementary Fig. 25).

In addition, it is well accepted that, for non-noble metal oxides-based CER catalysts, the activity decay is also a major issue in acidic electrolytes[1,7]. A highly crystalline and thick $Co_3O_4$ catalyst film prepared through calcination was used as a control[39], and we found that its current density quickly declined from 5.0 to 0.87 mA cm$^{-2}$ (Supplementary Fig. 26). This activity decay resulted from the leaching/dissolution of $Co^{2+}$ from the surface of $Co_3O_4$. Similarly, the noble metal oxides-based DSA and $RuO_2$ catalysts exhibited a fast descent in current density. In contrast, the current density of the $CoO_xCl_y$ catalyst deposited at 1.67 V increases to 10 mA cm$^{-2}$ and then stabilizes at about 15 mA cm$^{-2}$ over 500 h test, which leads to a downshift of potential ~ 300 mV at 10 mA cm$^{-2}$ relative to the initial CV activity (Supplementary Fig. 27). Meanwhile, the electrochemical stability of the $CoO_xCl_y$ at 250 mA cm$^{-2}_{geo}$ could sustain over 20 h (Supplementary Fig. 28). This catalyst system could even outperform the representative precious and non-precious metal oxide catalysts (Fig. 1e and Supplementary Table 1).

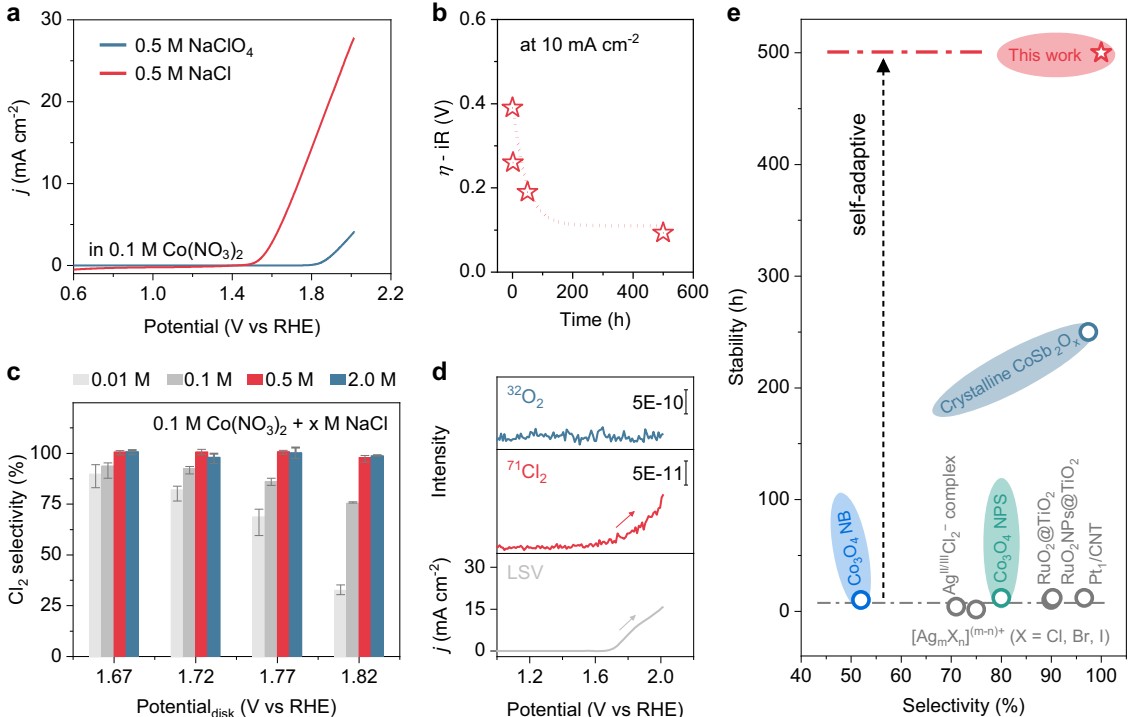

**Fig. 1 | The CER performance of amorphous $CoO_xCl_y$ catalysts. a** The LSV curve of $CoO_xCl_y$ and $CoO_x$ catalysts in 0.5 M NaCl and $NaClO_4$ at pH 2, respectively. The $CoO_xCl_y$ and $CoO_x$ catalysts were obtained after 2 h electrodeposition at 1.67 V in the electrolyte containing 0.1 M $Co^{2+}$ and 0.5 M $Cl^-$ or 0.5 M $ClO^-$ at pH 2, respectively. 1.67 V was selected based on the CER onset potential of the first potential scan. **b** The overpotential trend at 10 mA $cm^{-2}$ with increasing the electrodeposition time from the first potential scan to 500 h at 1.67 V in the electrolyte containing 0.1 M $Co^{2+}$ and 0.5 M $Cl^-$ at pH 2. The overpotential was iR corrected with 90% ohmic resistance. The electrolyte resistance was 46.0 Ω. **c** The $Cl_2$ selectivity of $CoO_xCl_y$ catalyst under different $Cl^-$ concentrations and anodic potentials during the deposition process at pH 2. The data were recorded based on a rotating disk electrode method at 1600 RPM. The error bars represent the standard deviation for triplicate measurements. **d** The LSV curves and corresponding DEMS signals during CER at pH 2 at 10 mV $s^{-1}$. **e** The performance comparison between different CER catalysts (the detailed data were summarized in Supplementary Table 1).

## $Cl^-$-induced deposition of amorphous $CoO_xCl_y$

To understand the catalyst deposition coupled CER process, we characterized the composition and structure of the deposited catalyst film. Compared to the $CoO_x$ film, the Co Kα and Cl Kα mapping of $CoO_xCl_y$ catalyst electrode by XRF show that the Co and Cl elements were distributed throughout the entire catalyst film, indicating the co-deposition of $Cl^-$ together with $Co^{2+}$ (Fig. 2a and Supplementary Fig. 29). The energy-dispersive X-ray spectroscopy (EDX) mappings confirmed the presence of the relatively even distribution of Co, Cl, and O for the formation of $CoO_xCl_y$ film even after 500 h stability test (Fig. 2b and Supplementary Fig. 30). The scanning electron microscope (SEM) images demonstrated the comparable dense surfaces of both $CoO_xCl_y$ and $CoO_x$ films (Supplementary Figs. 31, 32). The cross-section of $CoO_xCl_y$ film characterized by scanning transmission electron microscopy (STEM) showed a film thickness of about 600 nm after 40 h of electrodeposition at 1.67 V (Fig. 2c and Supplementary Fig. 33). The EDX mapping of the cross-section further exhibited the relatively homogenous distribution of Co and Cl elements (Supplementary Fig. 34). In addition, we characterized the structure of the electrodeposited catalyst films. X-ray diffraction (XRD) demonstrated that $CoO_xCl_y$ film was absent of characteristic diffraction peaks (Supplementary Fig. 35). The crystal structure of both $CoO_x$ and $CoO_xCl_y$ catalysts was further imaged by the HRTEM. The lattice fringe together with the selected area electron diffraction (SAED) pattern showed that $CoO_x$ film was composed of $CoO_x$ nanocrystals (Supplementary Fig. 36). In contrast, $CoO_xCl_y$ film exhibited an amorphous structure without lattice fringe and SAED rings (Fig. 2d). Even after 500 h of operation, the $CoO_xCl_y$ film still remained the amorphous structure (Supplementary Fig. 37). The amorphous nature was probably ascribed

to the $Cl^-$ co-deposited with $Co^{2+}$ in the catalyst as it may be induced by reconstruction or distortion relative to $CoO_x$[40].

## Suppression of Co oxidation state by $Cl^-$ in the $CoO_xCl_y$ catalyst

Further, compared with the $CoO_x$ film in the absence of $Cl^-$ (Supplementary Fig. 38), a peak at 197.6 eV for Cl $2p$ in $CoO_xCl_y$ film was observed, which was attributed to the Co-Cl bond (Fig. 2e)[25,41,42]. Analogous to Co (Supplementary Fig. 8 and 39a), the content of Cl increased with the electrodeposition time (Supplementary Fig. 39b and Supplementary Table 2). In addition, the chemical states of the as-prepared $CoO_xCl_y$ film were analyzed by XPS to study the role of $Cl^-$ inclusion. The Co $2p$ spectra can be well deconvoluted by the $Co^{2+}$ and $Co^{3+}$ components according to the reported binding energies[43,44] (Fig. 3a). The $Co^{3+}/Co^{2+}$ ratio in the $CoO_x$ film is higher than that in the $CoO_xCl_y$ film (Fig. 3b). Moreover, the Raman spectrum of $CoO_xCl_y$ film with peaks at 180.7, 456.2, 502.4, 591.5, and 660.1 $cm^{-1}$ belong to the 3 $F_{2g}$, 1 $E_g$, and 1 $A_{1g}$ vibrational modes of cobalt oxides (Fig. 3c)[45,46]. The blueshift of the vibration bands relative to those of $CoO_x$ film indicated the variation of Co local coordination owing to $Cl^-$ introduction, consistent with its relatively lower Co oxidation state (Fig. 3a)[40].

In situ X-ray absorption fine structure (XAFS) spectroscopy was further applied to elucidate the valence state of Co under CER conditions (Supplementary Fig. 40). The Co K-edge X-ray absorption near edge structure spectra (XANES) of $CoO_xCl_y$ and $CoO_x$ film were collected under anodic polarization in the 0.5 M NaCl and $NaClO_4$ electrolytes, respectively[47]. As shown in Fig. 3d and Supplementary Fig. 41, the anodic polarization induced a shift of the Co K-edge to a higher energy region for both the $CoO_xCl_y$ and $CoO_x$ film suggesting an increase in the Co oxidation state[44,48,49]. However, the $CoO_xCl_y$ film

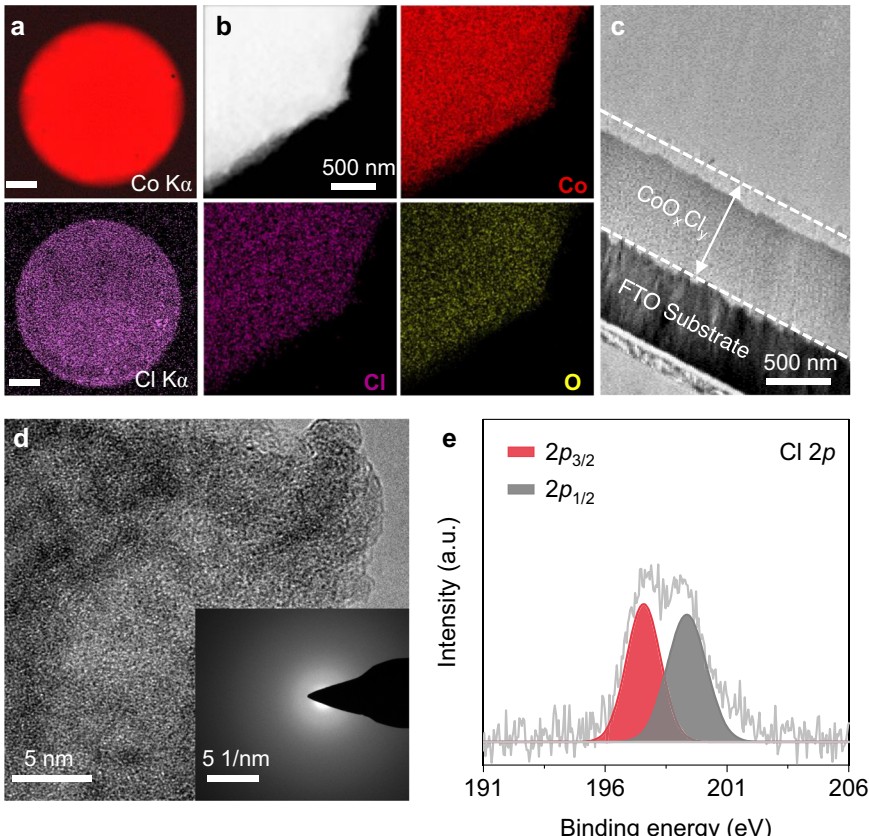

**Fig. 2 | The composition and structure of the CoO$_x$Cl$_y$ catalyst. a** The XRF Co Kα and Cl Kα mapping of the CoO$_x$Cl$_y$ film electrodeposited in the electrolyte containing 0.1 M Co$^{2+}$ and 0.5 M Cl$^-$ at 1.67 V for 10 h. The scale bar is 1 mm. **b** The EDX mapping of the CoO$_x$Cl$_y$ film after 40 h electrodeposition at 1.67 V. **c** The cross-section TEM images of the CoO$_x$Cl$_y$ film after 40 h electrodeposition at 1.67 V. **d** The HRTEM images and corresponding SAED patterns (inset) of the CoO$_x$Cl$_y$ film after 40 h electrodeposition at 1.67 V. **e** The Cl 2$p$ X-ray photoelectron spectroscopy (XPS) spectrum of the CoO$_x$Cl$_y$ film. This peak could be well fitted by 197.6 eV for Cl 2$p_{3/2}$ and 199.4 eV for Cl 2$p_{1/2}$, respectively[41,42].

underwent a larger shift (-0.83 eV) than that of the CoO$_x$ film (-0.41 eV) upon the potential increase from OCP to 2.02 V (Fig. 3e). Clearly, the CoO$_x$Cl$_y$ film showed lower Co K-edge absorption energy relative to that of CoO$_x$ film at all the test potentials. This suggested a relatively lower Co oxidation state in CoO$_x$Cl$_y$ film. Even at OCP, the Co valence state remains lower in the CoO$_x$Cl$_y$ relative to CoO$_x$, thus upon increasing the potential to 1.67 V, an obvious increase in the Co valence state allows CER to initiate[40]. Further, we analyzed the Fourier-transformed of k$^3$-weighted Co K-edge extended X-ray absorption fine structure (EXAFS) spectra of both in CoO$_x$ and CoO$_x$Cl$_y$ catalysts, which showed two major peaks associated with Co-O (~1.90 Å) and Co-Co (~2.85 Å) coordination (Supplementary Figs. 42, 43). An additional peak located around ~2.35 Å in the CoO$_x$Cl$_y$ catalyst was observed, indicating the presence of Co-Cl coordination (Supplementary Fig. 43). Compared to the CoO$_x$ catalyst, the CoO$_x$Cl$_y$ catalyst exhibited a lower fitted Co-O coordination number during the anodic polarization (Supplementary Tables 3, 4) and displayed a larger increase in Co-O coordination number relative to the CoO$_x$ catalyst, in coincidence with its significant degree of Co oxidation state variation[50,51].

To get a deeper understanding of the CER process together with the effect of Cl$^-$ introduction, in situ spectroelectrochemical UV-Vis was applied to track the deposition processes of CoO$_x$Cl$_y$ and CoO$_x$ catalysts (as illustrated in Fig. 4a and Supplementary Fig. 44). Figure 4b demonstrated the variation of spectral absorption of the CoO$_x$ film deposited in the absence of Cl$^-$ at 1.67 V. The absorption intensity increased rapidly within 30 min of deposition and several characteristic peaks appeared at A1-A4 corresponding to the charge transfer transitions[52,53]. The absorption centered at A2-A4 was assigned to the

higher oxidation state of Co coordinated with highly oxidized oxygen species in the oxygen evolution process[54–56]. Instead, A2-A4 absorbance bands were absent in CoO$_x$Cl$_y$ film, again suggesting a relatively lower oxidation state of Co than that in CoO$_x$ film (Fig. 4c). Moreover, the absorbance intensity of CoO$_x$Cl$_y$ film is almost two times that of the CoO$_x$ film. Such a phenomenon could be ascribed to the Cl$^-$ accelerated electrodeposition and CER process. To clarify this phenomenon, we tracked the absorption of electrodeposited CoO$_x$Cl$_y$ by stepwise increasing the Cl$^-$ concentration from 0 to 2.0 M (Supplementary Fig. 45). As expected, the A1 absorption intensity distinctly increased with the concentration of Cl$^-$ (Fig. 4d). When Cl$^-$ concentration was at 0.01 M, the A1 band intensity was relatively weaker while the A2-A4 appeared, indicating the existence of OER active phase analogous to that in CoO$_x$. These results together with in situ XANES analyses illuminate that the Cl$^-$ introduction promotes the electrodeposition process and improved the Cl$_2$ selectivity by suppressing the formation of highly oxidized Co sites required for OER.

## The role of Cl$^-$ on the CER process by DFT calculations

Subsequently, we applied the density functional theory (DFT) calculation to further clarify the effect of Cl$^-$ on Co valence states. Based on the above experiment results, we selected cobalt oxide with both octahedral and tetrahedral Co (Co$_{octahedral}$ /Co$_{tetrahedral}$ = 3) (Supplementary Fig. 46 and Supplementary Table 5), and constructed six possible explicit solution models based surfaces without and with Cl$^-$ in the catalyst (Supplementary Fig. 47). The negative formation energies suggested that Cl$^-$ was spontaneously introduced into the catalyst (Supplementary Note 4, Supplementary Table 6, and Supplementary

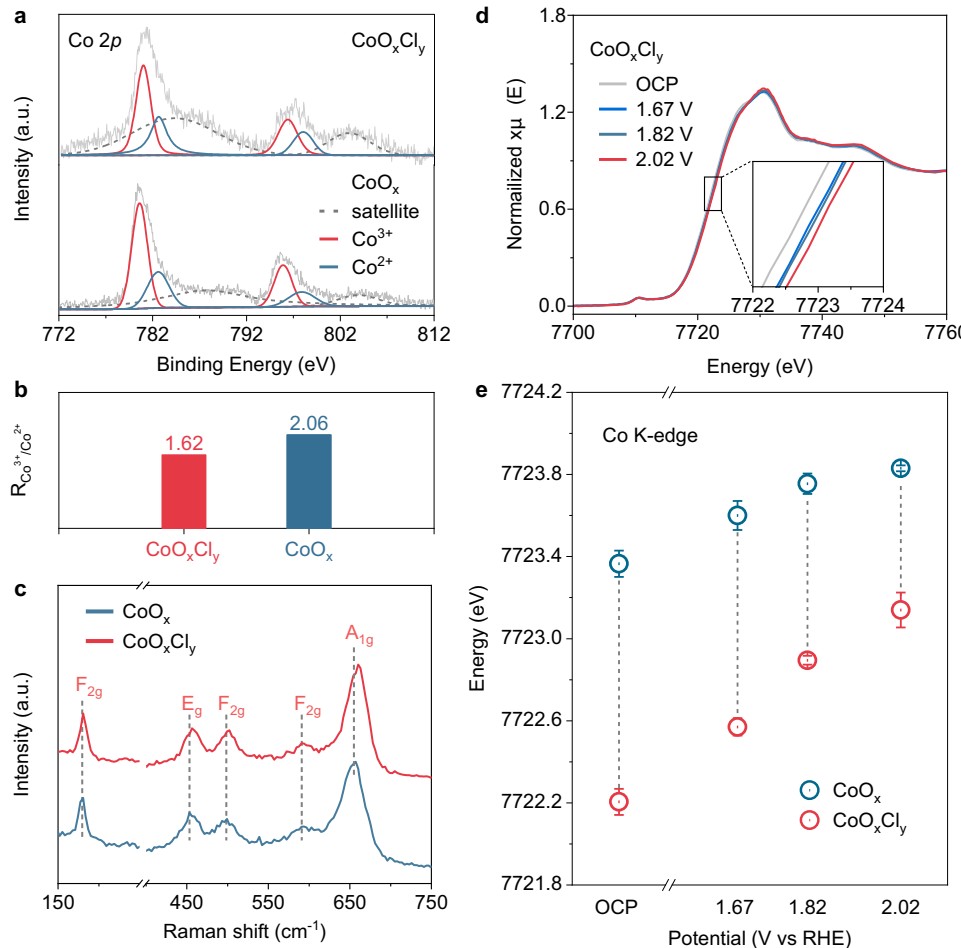

**Fig. 3 | The role of Cl⁻ on the Co oxidation state for the CoOₓClᵧ catalyst. a** The Co 2*p* XPS spectra of the CoOₓClᵧ and CoOₓ film after 2 h electrodeposition at 1.67 V, respectively. **b** The corresponding Co³⁺/Co²⁺ ratio of the CoOₓClᵧ and CoOₓ film. **c** The Raman spectra of the CoOₓClᵧ and CoOₓ films after 10 h electrodeposition at 1.67 V. **d** Co K-edge XANES at different applied potentials from the open-circuit condition to 2.02 V for CoOₓClᵧ film. **e** Co K-edge energy at different applied potentials for CoOₓ and CoOₓClᵧ film. The absorption energy (E₀) is obtained from the first maximum in the first-order derivative as the electronic vacancy. The error bars represent the standard deviation for triplicate measurements.

Fig. 48). We chose two models with lowest formation energy (Cl-doped-d and Cl-doped-e) to study the Co valence state in comparison with that of the Cl-free model (Supplementary Fig. 49). Because the surface atoms have lower coordination numbers than bulk atoms, surface Co sites are covered with water ligands. The density of electronic states (DOS) (Supplementary Fig. 50) shows that tetrahedra Co induces the defect states in the bandgap, attributed to the surface oxygens coordinative unsaturation and higher oxidation states of Co (Supplementary Fig. 51 and Supplementary Table 7). The defect states usually lead to significant optical absorption at the infrared and ultraviolet-visible regions[57], which may correspond to the A2-A4 absorbance of CoOₓ film. When the surface oxygen atoms are substituted by Cl⁻, the defect states disappeared, in line with the absence of A2-A4 absorbance bands of CoOₓClᵧ film. Moreover, the hole states near the Fermi level appear to suggest Cl⁻ in the catalyst enhanced electrical conductivity with reduced charge transfer resistance (Supplementary Fig. 13). As shown in Supplementary Figs. 52, 53 and Supplementary Tables 8, 9, the calculated Bader charge and bond length suggested that Cl⁻ introduction could increase the Co electronic charge and decrease the valence state, consistent with in situ UV-Vis and XANES results.

To gain an in-depth understanding of the high Cl₂ selectivity of CoOₓClᵧ, we further studied the influence of Cl⁻ introduction on OER and CER. We first constructed the OER and CER reaction free-energy diagrams as a function of electrode potential (U) at pH 2 and 0.5 M Cl⁻ based on the thermodynamically stable structures of various adsorbates[10,58] (i.e. Cl*, ClO*, H₂O*, HO*, O* and HOO*) on Co sites (Supplementary Note 5). Without Cl⁻ in the catalyst, Cl* is favorable at U < 1.4 V, which transferred to Cl-O* at a higher potential (Fig. 5a). On the other hand, in the presence of Cl in the catalyst, OH* became more favorable at U < 1.4 V (Fig. 5b and Supplementary Fig. 54). Hence, we evaluated the influence of pH and Cl⁻ concentrations on the reaction pathways (Supplementary Fig. 55). The results indicated that even with very low Cl⁻ concentrations, CER is predominant at pH 2 on both Cl-free and Cl-containing catalysts.

To further understand the high selectivity of Cl₂ to O₂, we compared the reaction free-energy diagrams between OER and CER (Fig. 5c, d and Supplementary Note 6). For CER, we consider the mechanism that proceeded through the Volmer (O*/* + Cl⁻ → ClO*/Cl* + e⁻) and the subsequent Heyrovsky step (ClO*/Cl* + Cl⁻ → Cl₂ + e⁻), where the Cl⁻ directly adsorbed on the active site. On the Cl-free catalyst, the theoretical overpotentials for the OER and CER were 0.63 V and 0.39 V, respectively (Fig. 5c, d, Supplementary Figs. 56–59, and Supplementary Tables 10, 11). On the Cl-containing catalyst, the theoretical overpotential for both the OER and CER were reduced to 0.41 V and 0.2 V, respectively (Fig. 5c, d, Supplementary Figs. 60–63, and Supplementary Tables 12, 13). Even though the theoretical potential of 1.36 V for CER is 0.13 V higher than the theoretical

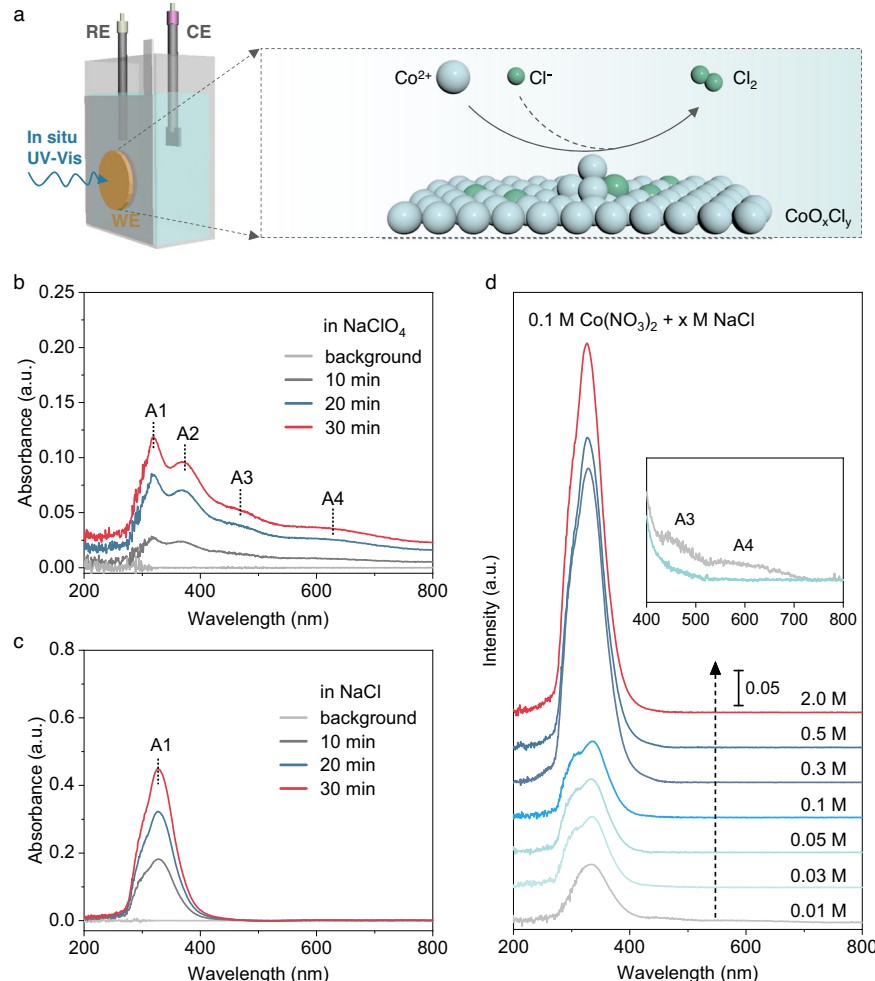

**Fig. 4 | In situ UV-Vis spectra for tracking the electrodeposition process of the CoOxCly and CoOx film on FTO. a** The schematics illustrate the in situ spectroelectrochemical UV-Vis tracking of the catalyst deposition process (left). The schematic diagram for CER on the CoOxCly electrode (right). **b** The UV-Vis absorption spectra of the deposited CoOx film in the electrolyte containing 0.1 M Co$^{2+}$ and 0.5 M ClO$_4^-$ at 1.67 V. **c** The UV-Vis absorption spectra of the deposited CoOxCly film in the electrolyte containing 0.1 M Co$^{2+}$ and 0.5 M Cl$^-$ at 1.67 V. **d** The UV-Vis absorption spectra of the deposited CoOxCly film upon increasing the Cl$^-$ concentration from 0.01 to 2.0 M. Inset shows the absorption spectra around the A2-A4 range for the CoOxCly electrodeposited in 0.01 M Cl$^-$ electrolyte.

potential of 1.23 V for OER, such results still suggest that CER is more favorable than OER on the catalysts with and without Cl. However, the presence of Cl$^-$ in the catalysts can accelerate the catalysis reaction. In addition, there is a transition of the active site from Co* to the Co-O* for CER with the Cl$^-$ introduction into the catalyst (Fig. 5e). The same trend was observed for other Cl-containing models (Supplementary Figs. 64–67, and Supplementary Tables 14, 15).

In this work, we present a highly selective and robust CER system based on an amorphous CoOxCly electrocatalyst electrodeposited in an acidic electrolyte containing both Co$^{2+}$ and Cl$^-$ ions. Notably, this work presents the first case in that Cl$^-$ ions work as a promoter for catalyst deposition and CER. Experimental results demonstrate that ~100% CER selectivity has been acquired and in situ spectroscopic studies together with DFT theoretical calculations show that the introduction of Cl$^-$ into the catalyst suppresses the formation of highly oxidized Co sites required for water oxidation while allowing for the active site transition from Co* to Co-O* for CER with enhanced CER selectivity via the Volmer-Heyrovsky pathway. This work provides a new avenue to stabilize the non-precious metal oxides for anodically electrocatalytic reactions in strong acids and we would suggest in situ characterize the catalyst system with cautions before we categorize a soluble species or molecule as a homogeneous catalyst.

## Methods

### Chemicals

All chemicals used in this work were commercially available. Cobalt nitrate hexahydrate (Co(NO$_3$)$_2$·6H$_2$O, 99.99%), cobalt chloride hexahydrate (CoCl$_2$·6H$_2$O, 99.99%), ferric nitrate nonahydrate (Fe(NO$_3$)$_3$·9H$_2$O, 99.99%), manganese nitrate tetrahydrate (Mn(NO$_3$)$_2$·4H$_2$O, 99.99%), potassium ferricyanide trihydrate (K$_3$Fe(CN)$_6$·3H$_2$O, 99.99%), urea (CO(NH$_2$)$_2$, 99.999%), sodium chloride (NaCl, 99.99%), ammonium fluoride (NH$_4$F, 99.99%), perchloric acid (HClO$_4$, 70–72%), and anhydrous ethanol (C$_2$H$_6$O, 99.8%) were purchased from Sigma-Aldrich. Cobalt sulfate heptahydrate (CoSO$_4$·7H$_2$O, 99.99%), nickel nitrate hexahydrate (Ni(NO$_3$)$_2$·6H$_2$O, 99.99%), copper nitrate trihydrate (Cu(NO$_3$)$_2$·3H$_2$O, 99.99%), sodium perchlorate monohydrate (NaClO$_4$·H$_2$O, 99.99%), and cobalt acetate tetrahydrate (Co(acetate)$_2$·4H$_2$O, 99.9%) were purchased from Aladdin. Hydrochloric acid (HCl, 31%), nitric acid (HNO$_3$, 65–68%), sulfuric acid (H$_2$SO$_4$, 95–98%), and acetone (C$_3$H$_6$O, 99.9%) were purchased from KESHI. Commercial ruthenium oxide (RuO$_2$, Ru > 75%) and iridium oxide (IrO$_2$, Ir > 85%) were purchased from Adamas. In all experiments, ultrapure deionized water (Milli-Q, 18.2 MΩ cm$^{-1}$) was used for the electrolyte preparation. The clean fluorine-doped tin oxide (FTO) with resistance <15 ohm/sq was used as a substrate for electrodeposition. All chemicals were used without further purification.

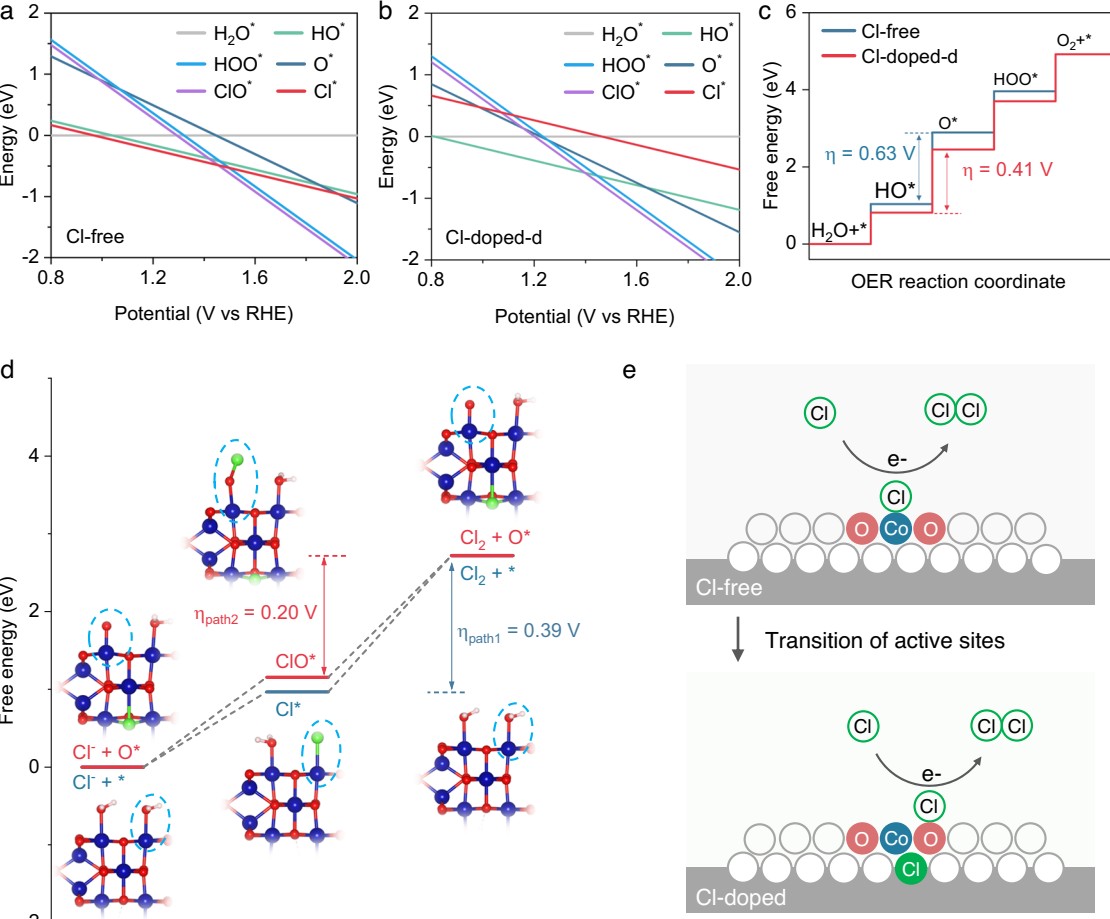

**Fig. 5 | DFT calculations to understand the role of $Cl^-$ on the reaction mechanism. a** The surface phase diagram of the Cl-free catalyst model. **b** The surface phase diagram of the Cl-containing catalyst model. **c** The reaction free energy diagram of OER on the Co sites of the Cl-free and Cl-containing catalysts, respectively. **d** The reaction free energy diagrams of CER on the Co sites of the Cl-free and Cl-containing catalysts, respectively. Insets: corresponding computational models for each CER step. **e** The schematic illustration of the role of $Cl^-$ introduction on the CER reaction active sites.

## In situ electrodeposition of $CoO_xCl_y$ catalyst

The electrolyte for the working electrode side was prepared by mixing 0.1 M $Co^{2+}$ salts and 0.5 M NaCl in 40 mL of deionized water. The pH of the electrolyte was adjusted to 2 using $HClO_4$. Typically, the $CoO_xCl_y$ film was in situ electrodeposited on the FTO in the Co salt-NaCl electrolyte at 1.67 V at 25 °C. The electrochemical tests were conducted in a custom-designed PEEK cell. The Pt counter electrode was separated by an N117 Nafion membrane with a thickness of 183 um and size of 4 × 4 cm and was immersed in 0.5 M NaCl (pH 2) as well, to avoid the corrosion and the cathodic deposition of Co-based film. Before the experiment, the membrane was inserted in 5% hydrogen peroxide soaking for 60 min, deionized water for 30 min, 5% diluted sulfuric acid for 60 min, and deionized water for 30 min at around 80 °C.

## Synthesis of $Co_3O_4$ film

Typically, the $Co_3O_4$ electrode was prepared by a simple hydrothermal method following a previous report[39]. Specifically, 0.233 g $Co(NO_3)_2 \cdot 6H_2O$, 0.07 g of $NH_4F$, and 0.240 g of $CO(NH_2)_2$ were dissolved in 100 mL deionized water under stirring for 30 min. Then, the obtained solution was transferred to a 50 mL Teflon-lined autoclave with an FTO inside as substrate. The reaction was conducted at 120 °C for 10 h and then cooled to room temperature. The obtained FTO substrates were washed with deionized water, dried at 60 °C overnight, and annealed in air at 350 °C for 2 h with a heating rate of 2 °C $min^{-1}$.

## Synthesis of $RuO_2$ and DSA catalysts

To prepare the $RuO_2$ film electrode, 5 mg commercial $RuO_2$ was dissolved into 0.95 mL ethanol by adding 0.05 mL 5 wt% nafion117 solution under ultrasonic treatment for 60 min. 10 uL of this suspension was dropped onto the clean FTO (the efficient contact area is 0.283 $cm^2$) for the formation of a catalyst film. The DSA catalyst film was prepared based on the commercial protocol on Ti (the ratio of Ir/Ru is 2).

## Electrochemical characterizations

The CER tests were conducted based on a standard three-electrode with a computer-controlled electrochemical workstation (Bio-Logic SP −200). FTO (efficient contact area: 0.283 $cm^2$), platinum mesh, and Hg/$Hg_2SO_4$ electrode were used as the working electrode, counter electrode, and reference electrode, respectively. A custom-designed PEEK cell with a quartz window was used as the electrochemical cell. All experiments were carried out in a temperature-controllable box, and the temperature was kept at 25 °C if not specially indicated. Before electrochemical experiments, the reference electrodes were calibrated by a standard reversible hydrogen (RHE) electrode (PHYCHEMI) in the corresponding electrolytes after 60 min Ar bubbling. For the working electrode, magnetic stirring at 1500 RPM was applied to remove chlorine bubbles during the electrochemical tests. All the measured potentials were calibrated with a reversible hydrogen electrode (RHE) as the reference. The pH values of the electrolyte were measured through an Ohaus Starter 2100 pH meter with temperature calibration.

For the electrochemical tests at different NaCl concentrations, the $NaClO_4 \cdot H_2O$ was added for compensation to maintain a constant ionic strength[8]. The ohmic resistance (R) for iR corrections was determined by electrochemical impedance spectroscopy (EIS), and the recorded ohmic resistance was around 46.0 Ω, as shown in Supplementary Fig. 13. The overpotential in Fig. 1b and potential in Supplementary Fig. 28 were iR corrected with 90% ohmic resistance, while the other data were not iR-corrected. Electrochemical impedance spectroscopy (EIS) was measured at 1.67 V with a frequency scan range from 100 kHz to 10 mHz, and the amplitude of the sinusoidal wave was 10 mV.

## In situ X-ray Absorption Fine Structure (XAFS) measurements

The XAFS measurements were recorded at the BL11B beamline of the Shanghai Synchrotron Radiation Facility (SSRF). The beam current of the storage ring was 220 mA in a top-up mode and the Si (111) double-crystal monochromator was applied to monochromatize the incident photons, with an energy resolution ΔE/E ~ 2×10$^{-4}$. The collected spot size of the sample was ~200 μm × 250 μm (H × V). And the Co foil was used to calibrate the position of the absorption edge ($E_0$). All XAFS spectra were collected in fluorescence mode. The $E_0$ was obtained from the largest peak in the 1$^{st}$ derivative XANES according to the previous work[47,59]. The $CoO_xCl_y$ and $CoO_x$ films were polarized at different anodic potentials in a three-electrode system, respectively. After the corresponding current density reaches a steady state, the XAFS data was collected at the indicated potentials. All XAFS data were processed by the ATHENA and ARTEMIS modules implemented in the IFEFFIT software package.

## In situ UV-Vis

The in situ UV-Vis spectroscopy was implemented on a QE Pro UV-Visible spectrometer (Ocean Optics) equipped with an HL-2000 light source and a DH-2000-BAL source. A fiber-optic cable (100 μm fiber core diameter) was used to direct the light from the light source through the sample to the detector. The UV-Vis spectra were recorded in the different electrolytes as a function of electrolysis times at 1.67 V. The FTO (area: 0.283 cm$^2$), Pt electrode, and $Hg/Hg_2SO_4$ were used as the working electrode, the counter electrode, and the reference electrode, respectively. The spectrum of the unpolarized cleaned FTO immersed in the same electrolyte was recorded for the background subtraction of the UV-Vis spectra of the deposited $CoO_xCl_y$ and $CoO_x$ catalysts.

## Online differential electrochemical mass spectrometry (DEMS)

The online DEMS experiments were implemented with the chamber containing the quadrupole mass spectrometer (MS) with a Faraday-SEM detector array (Hiden Analytical, HPR-40), which allows the isolation of the ion source from the electrochemical cell by forming a small pre-chamber[37,38]. For the DEMS measurements, a custom gas-tight electrochemical H-cell with a conventional three-electrode arrangement was employed for the electrochemical experiments. In this experiment, a Nafion membrane was used to separate the cathode and anode. The acquisition probe is vertically attached to the working electrode where the volatile species pass through a porous Teflon membrane into the vacuum inlet of the mass spectrometer. For the DEMS measurements, the multiplier mode was adopted. The LSV and mass spectrometric cyclic voltammograms (MSCVs) were acquired in electrolytes containing 0.1 M Co$^{2+}$ with 0.5 M NaClO$_4$ or 0.5 M NaCl, respectively. The MSCVs for the concerned species ($^{70}Cl_2$: m/z = 70, $^{1}H^{16}O^{35}Cl$: m/z = 52, $^{35}Cl^{32}O_2$: m/z = 67, $^{35}Cl^{16}O$: m/z = 51 and $^{32}O_2$: m/z = 32) were recorded. Before the tests, the electrolyte was purged with Ar (99.999%) over 1 h to remove the dissolved oxygen.

## Materials characterizations

The X-ray fluorescence method based on Co and Cl content analyses was conducted on a Bruker M4 Tornado instrument with Rh target for X-ray generation. The catalyst morphology of the samples was characterized by a SU8010 field emission scanning electron microscope (SEM). The cross-section of the catalyst was characterized by a transmission electron microscope (JEOL JEM2100F) operating at 200 keV in both TEM and STEM modes. The electron beam spot size is 1.0 nm. An EDX detector (Oxford Instruments) was equipped for the element analysis. The TEM sample for cross-section observation was prepared by cutting the electrodeposited FTO electrode into two pieces, of which the cross-section was firstly glued by the epoxy resin and then processed through Ar$^+$ ion-milling and polishing (PIPS II, GATAN) at grazing incidence (<5°). The X-Ray Diffraction (XRD) measurement of $CoO_xCl_y$ catalyst was conducted on Bruker D8 ADVANCE A25X. The high-resolution TEM (HRTEM) images of $CoO_xCl_y$ and $CoO_x$ catalysts were taken on an FEI-Tecnai G2 microscope. Moreover, the element mapping and selected area electron diffraction (SAED) were performed on an FEI TECNAI G2 F30 field emission transmission electron microscope. The X-ray photoelectron spectroscopy analyses were carried out using a Thermo Scientific K-Alpha system to evaluate the composition and chemical states of $CoO_xCl_y$ and $CoO_x$ catalysts. Al Kα X-ray source with a power of 250 W was used as the X-ray source and the binding energies were calibrated by referring to the C 1s peak (284.8 eV) of the adventitious carbon. The Raman was carried out with an XploRA PLUS Raman spectrometer (Horiba) equipped with a 532 nm He-Ne laser as the excitation source.

## Rotating ring-disk electrode detection of selectivity Cl$_2$ evolution

The Cl$_2$ selectivity for CER was analyzed based on the RRDE system in which the disk electrode generates Cl$_2$ while the Pt ring electrode monitors Cl$_2$ concentration through its electrochemical reduction current. The potential of the Pt ring was fixed at 0.95 V to avoid the current response from O$_2$ reduction. Before each measurement for Cl$_2$ quantification, the Pt ring-disk electrode was successively polished with aqueous suspensions of 1.0 μm, 0.3 μm, and 0.05 μm alumina. It was then rinsed with deionized water, ultrasonically cleaned in ethanol for 15 s, and dried. Later on, the Pt ring was further electropolished in 0.5 M H$_2$SO$_4$ by 30 CV scans between -0.1 and 1.7 V at a rate of 500 mV s$^{-1}$. The chronoamperometry method was applied for the Cl$_2$ selectivity quantification. During the Cl$_2$ selectivity measurements, the potentiostatic electrolysis was carried out for 5 min at 1600 RPM at different applied disk potentials[35]. Each experiment was repeated three times with an interval of 5 min in an Ar-saturated electrolyte at pH 2 containing 0.1 M Co(NO$_3$)$_2$ and NaCl with different concentrations.

The CER current density ($j_{CER}$) on the disk electrode can be calculated by the following equation:

$$j_{CER} = |j_R / N_l| = |i_R/A \times N_l| \tag{1}$$

Where $j_R$ is the current density measured on the ring, N$_l$ is the collection efficiency (0.37), and A is the disk electrode area (0.196 cm$^2$).

The OER current density ($j_{OER}$) can be calculated by the following equation:

$$j_{OER} = j_D - j_{CER} = j_D - |j_R/N_l| = j_D - |i_R/A \times N_l| \tag{2}$$

Where $j_D$ is the current density measured on the disk.

The Cl$_2$ selectivity was calculated by the following equation:

$$\begin{aligned} Cl_2 \; selectivity(\%) &= 100 \times 2 \times j_{CER}/(j_D + j_{CER}) \\ &= 100 \times 2 \times |j_R/N_l|/(j_D + |j_R/N_l|) \end{aligned} \tag{3}$$

## Computation details

The first-principles calculations based on density functional theory (DFT) were carried out within the projected augmented wave

method, as implemented in the Vienna Ab-initio Simulation Package (VASP). The $CoO_xCl_y$ film catalyst models were made by cleaving along the (100) facet of $Co_3O_4$ which contained two tetrahedrons and six octahedrons and were solved by explicit solution models. To generate highly accurate electrochemical stability diagrams, we employed a recently developed approach, which included the use of a Hubbard U term, a van der Waals functional (optPBE), and the use of spin polarization for the calculations. The U-value applied for d-orbitals of Co is taken as 3.50 eV. A cutoff energy of 500 eV was used for the plane wave expansion at all calculations of structure relaxations, vibrational frequencies, and single point energy. Monkhorst-Pack ($4 \times 4 \times 1$) k-point grids were used for Brillouin zone integration. The equilibrium structures were obtained when the maximum atomic forces were smaller than 0.01 eV/Å and when the total energy convergence of $10^{-6}$ eV was achieved for the electronic self-consistent field loop. The thermodynamic energy correction used in the free energy calculations was listed in Supplementary Table 6. The detailed descriptions of formation energy, OER, and CER reaction free energies calculations have been discussed in Supplementary Notes 4–6.

## Data availability
All relevant data are provided in this article and its Supplementary Information.

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

## Acknowledgements

C.C. acknowledges the funding support from the Natural Science Foundation of China (22072013, 22372027). C.L. acknowledges the financial support from the Natural Science Foundation of China (22202034) and China Postdoctoral Science Foundation (2022M720657). J.L. acknowledges the financial support from China Scholarships Council (No.202008455017). A.M. and M.V. acknowledge the financial support from the Slovenian Research Agency (research core funding No. P2-0412 and project No. J2–2498). The XAFS beam time was granted by BL11B end station of Shanghai Synchrotron Radiation Facility, Chinese Academy of Sciences. The staff members of BL11B are acknowledged for their support in measurements and data analyses.

## Author contributions

C.C. led the project. M.X. carried out the experiments. Q.W. repeated the experiments. C.L., M.X., J.L. (Jiong Li), and Q.W. finished the XAFS measurements. L.L. and M.X. proceed with the DEMS. A.M. and M.V. characterized STEM. J.L. (Junwu Liang), Z.Z., J.Z., L.Z., and R.K. conducted the density functional theory calculation. M.X. and C.C. wrote the manuscript. All authors commented on the manuscript.

## Competing interests

The authors declare no competing interests.
