## [Peer Review file · Nature Communications]

REVIEWER COMMENTS

Reviewer #1 (Remarks to the Author):

Xiao and his colleagues synthesized CoOxCly electrocatalyst in situ using Co and Cl salts and investigated its electrocatalytic activity towards the chlorine evolution reaction (CER) in this manuscript. They analyzed the physical and electrocatalytic properties in detail using RRDE, DEMS, XRF, and XPS. Additionally, they performed advanced in situ spectroscopic measurements using XANES and UV-Vis to reveal the fundamental mechanisms, which were further supported by computational approaches. The manuscript is well-written and highly readable, but it requires revision to address the following concerns:

Although the authors initially screened various transition metal precursors for in situ active site formation and subsequent CER activity, they found that only Co showed a large difference in onset potential and CER activity compared to those in the Cl-free electrolyte. However, Fe also showed similar trends to Co, as shown in Fig. S1. Thus, more comprehensive studies are required: the effects of Fe and Cl concentrations on CER activity, DEMS, stability, and other advanced analyses. A discussion of the catalytic ability of Fe would be beneficial to future researchers in this field.

The validity of the DEMS results is questionable, as the polarization curve measured during DEMS analysis showed significantly lower catalytic activity than that measured using a conventional three-electrode setup. The authors should clarify the reason for this discrepancy. Additionally, the evolution of the O₂ signal during DEMS analysis in the Cl-free electrolyte was unclear. The authors argued that O₂ evolved exclusively in these conditions, but no clear O₂ signal was observed. The reviewer assumed that the authors may have measured the signal in multiplier mode, which can induce artifacts when analytes are abundant and saturate the DEMS signal. Alternatively, the DEMS may not detect minute amounts of analyte. In this case, the multiplier mode may work properly. The absence of an O₂ signal in the Cl-free electrolyte is unusual, considering the intensity of all DEMS signals.

The authors did not clearly state the operating conditions for CER. Did they use a NaCl electrolyte with Co ions? If so, how did the dynamic equilibrium between Co deposition and dissolution affect CER electrocatalysis? Additionally, is this strategy industrially feasible? Alternatively, if the authors used a NaCl electrolyte free from Co ions, how did the transition metal oxide survive in such a highly corrosive environment?

The manuscript lacks physical characterizations after the stability test.

Overall, the paper shows great potential, but it needs to address these concerns before publication in Nature Communications.

Reviewer #2 (Remarks to the Author):

This work reports an amorphous CoOxCly catalyst that was in situ synthesized in an acidic saline electrolyte, using Co²⁺ and Cl⁻ ions to adapt to the specific electrochemical conditions. The catalyst demonstrated an impressive CER selectivity of ~100% with an overpotential of approximately 0.1 V at a current density of 10 mA cm⁻². Notably, the catalyst remained stable for 500 hours. The addition of Cl⁻ to the catalyst was found to hinder the formation of highly-oxidized Co sites, which are typically required for water oxidation, while allowing for the transition from Co* to Co-O* as the active site for CER, ultimately leading to an enhanced CER selectivity. However, I am concerning the actual atomic structure of the active site and CER performance of the CoOxCly catalyst. Although the authors employed in situ XAFS to study the valence state of Co under different CER conditions, there is a lack of evidence for the active site atomic structure. The CoOxCly catalyst was prepared by in situ deposition under CER reaction potential to improve its stability. However, stability test conditions of the reported catalyst are not consistent, e.g., crystalline CoSb₂O₇ stability was measured at a current density of 100 mA cm⁻², while RuO₂@TiO₂ stability was measured at a current density of 250 mA cm⁻², making it difficult to compare. In addition, there is still a gap between the activity of CoOxCly catalyst and that of noble or non-noble metal catalysts reported in the literature, especially in terms of ultimate current density. In general, this study still needs further refinement that falls short of Nature Communications' requirements.

Thus, this reviewer does not suggest the publication of this manuscript. There are some other comments on the manuscript that I hope will contribute to the successful publication of this work in the future.

1. It is recommended to incorporate the LSV test following catalyst activation. Furthermore, a comparison of the activity, selectivity, and stability of the catalyst with RuO₂ and DSA is advisable.
2. The Co K α and Cl K α mapping of CoOxCly catalyst electrodes by XRF shows that Co and Cl elements are distributed throughout the catalyst film, but their concentration varies across the film. Figure 2a shows that Co is more concentrated in the middle, while Cl is concentrated at the edge. Is this reasonable?
3. The authors used XRF and XPS to characterize the changes in Cl content as "Analogous to Co (Supplementary Fig. 8 and 26a), the content of Cl increased with the electrodeposition time (Supplementary Fig. 26b)". Can the authors give the magnitude or range of x and y in CoOxCly? This is informative for the reasonableness of the theoretical calculation modeling.
4. The amorphous structure of CoOxCly film in the paper is only demonstrated by HRTEM images and corresponding SAED patterns. Can other characterization techniques, such as X-ray diffraction (XRD), be

used in conjunction with other methods to increase the level of confidence in the amorphous structure of CoOxCly catalysts, particularly after activation?

5. The region chosen in Figure 3d may be not reasonable, and it is suggested to reconfirm the absorption energy (E₀), obtained from the first maximum in the first order derivative as the electronic vacancy, before determining the oxidation state of Co. It is suggested to refer to *Angew. Chem. Int. Ed.* 2022, e202209486 and *J. Am. Chem. Soc.* 2017, 139, 12076-12083. In addition, as seen from the intensities of the white-line peaks in Figure 3d, the CoOxCly valence at 1.67 V is the lowest, which is not consistent with the authors' conclusion.

6. The equilibrium potential of CER is correlated with temperature, chloride ion activity and pH, and the authors need to reconfirm the equilibrium potential in this paper.

Reviewer #3 (Remarks to the Author):

This paper by Xiao et al. addresses the very challenging issue that non-precious catalysts cannot stand for selective chlorine evolution reaction in the very corrosive and strong acid. The authors proposed an idea that allows the catalyst deposition to adapt to the hostile electrochemical conditions, with an unexpected amorphous CoOxCly catalyst in situ deposited on electrodes in acidic brine containing Co²⁺ ions. This resultant catalyst presents 100% selectivity and long-term stability over 500 hours. In combination in-situ spectroscopies with theoretical calculations, the authors find the key role of reactant Cl⁻ itself as a promoter for catalyst deposition and chlorine evolution reaction. Overall, the authors have done a nice work since it can be a reference for acidic sea water splitting that produces pure chlorine gas rather than oxygen on the anodes. Before I could recommend it for publication in *Nature Communications*, the authors should address a few issues that I have highlighted below.

Main text:

(1), page 5 Fig. 1b – After a certain time of electrodeposition process, the catalyst seems to approach to the lowest overpotential. The authors may specify which part is the adaptive deposition process and suggest the possibility to accelerate this process?

(2), page 5 Fig. 1c – The authors applied the RRDE to measure the Cl₂ selectivity at different potentials in acidic electrolyte containing Co²⁺. A potential of 0.95 V was applied on Pt ring. How to exclude the possible deposition of Co on Pt ring at this potential?

(3), page 6, lines 142 - 143 and page S21, Supplementary Fig. 19 – the initial current density $\sim 5.0 \text{ mA cm}^{-2}$ of CoOxCly at 1.67 V before the deposition process is closed to Co₃O₄. It seems that there is a very rapid deposition of CoOxCly. What are the mass loadings of Co₃O₄ and CoOxCly catalysts?

(4), page 7, line 161 – there is a typo - “Supplementary Figs. 23” should be “Supplementary Fig. 23”.

(5), page 8, Fig. 2a – the color bar is missing.

(6), page 13, line 278; page 14, line 280 – better using “theoretical overpotential” in DFT calculations.

Supplementary information:

(7), page S1, lines 30-31 – “The electrolyte at pH = 2 containing 0.001 M indicated transition metal salts and 0.5 M NaCl (for CER)/NaClO₄ (for OER)”. The description is confusing.

(8), page S12-S17, Supplementary Fig. 12-15 – the error bar is confusing. The calculated Cl₂ selectivity at the corresponding current density/potential should be constant, the error bars obtained by repeated measurements have already been shown in Fig. 1c.

(9), page S23, Supplementary Fig. 20 – the scale bar is missing.

(10), page S23, Supplementary Fig. 20b – the authors may provide the contrasting background of Cl K α mapping.

(11), page S32, Supplementary Table 2 – there are two typos. “Co³⁺ and Co⁴⁺” should be “Co²⁺ and Co³⁺”.

Reviewer #1 (Remarks to the Author):

Xiao and his colleagues synthesized CoOxCl_y electrocatalyst in situ using Co and Cl salts and investigated its electrocatalytic activity towards the chlorine evolution reaction (CER) in this manuscript. They analyzed the physical and electrocatalytic properties in detail using RRDE, DEMS, XRF, and XPS. Additionally, they performed advanced in situ spectroscopic measurements using XANES and UV-Vis to reveal the fundamental mechanisms, which were further supported by computational approaches. The manuscript is well-written and highly readable, but it requires revision to address the following concerns:

Responses 1:

- We truly appreciate your positive comments on our work. The comments and suggestions are quite helpful, please find our response and revision as follows.

Although the authors initially screened various transition metal precursors for in situ active site formation and subsequent CER activity, they found that only Co showed a large difference in onset potential and CER activity compared to those in the Cl-free electrolyte. However, Fe also showed similar trends to Co, as shown in Fig. S1. Thus, more comprehensive studies are required: the effects of Fe and Cl concentrations on CER activity, DEMS, stability, and other advanced analyses. A discussion of the catalytic ability of Fe would be beneficial to future researchers in this field.

Response 2:

- Thanks very much for the suggestions from the referee. We have supplemented a detailed study to investigate the effect of Fe and Cl concentrations on CER. As shown in Fig. R1 a-b, the Fe-catalyst only demonstrated an initial activity for CER in NaCl relative to the NaClO₄ electrolyte, yet the CER current densities did not significantly increase with increasing either Fe³⁺ or Cl⁻ concentrations (Fig. R1). This is because the continuous electrodeposition of the Fe-catalyst film is not favorable.
- In contrast to the Co-catalyst, the Fe-catalyst has two limitations. First, we noticed the dominant hydrolysis of Fe ions even at pH 2 (Fig. R2), resulting in the decrease of soluble Fe ions. Second, we showed that a very limited amount of Fe could be deposited on the electrode (Fig. R3). Thus, no obvious increase in current density was observed, and the onset potential remained almost unchanged during the 2-hour operation (Fig. R4). Due to the limited number of deposited Fe active sites, the generated Cl₂ was little (Fig. R5).

Fig. R1 The CV curves for CER and OER under different Fe^{3+} and Cl^- concentrations. The CV curves for CER containing different concentrations of $\text{Fe}(\text{NO}_3)_3$ in (a) 0.5 M NaCl electrolyte and (b) 0.5 M NaClO_4 . The CV curves for CER in 0.1 M $\text{Fe}(\text{NO}_3)_3$ electrolytes containing different concentrations of (c) NaCl and (d) NaClO_4 . All measurements were conducted at pH = 2 and CV curves were recorded at 10 mV s^{-1} .

Fig. R2 The hydrolysis of iron. Color of 1.0 M $\text{Fe}(\text{NO}_3)_3$ (I), 0.1 M $\text{Fe}(\text{NO}_3)_3$ (II), 0.01 M $\text{Fe}(\text{NO}_3)_3$ (III), and 0.001 M $\text{Fe}(\text{NO}_3)_3$ (IV) solutions in 0.5 M NaCl solution. The light beam shows the hydrolysis of Fe^{3+} for the formation of insoluble colloids.

Fig. R3 The X-ray fluorescence (XRF) Fe K α mapping of the catalyst film electrodeposited in the electrolyte containing 0.1 M Fe^{3+} and 1.0 M Cl^- at $1.83 \text{ V}_{\text{RHE}}$ for 2.0 h. XRF mappings showed that there was no significant deposition of Fe.

Fig. R4 The CER activity trend with the operating time. **(a)** 2 h of potentiostatic operation and **(b)** CV curves of the electrode before and after 2 h potentiostatic operation at 1.83 V_{RHE} in 0.1 M Fe(NO₃)₃ + 1.0 M NaCl electrolyte.

Fig. R5 The determination of Cl₂ generation by KI test papers during the LSV. The wet potassium iodide-starch test paper was positioned in the headspace of a reaction cell during the anodic polarization process. The color change from white to blue-purple suggested the oxidation of I⁻ to I₂ by the electrochemically generated Cl₂.

The supplemented discussion concerning the CER activity of the Fe catalyst was added to the *main text* and the *supplementary information* as follows:

“However, although the Fe-catalyst displayed initial CER activity, the subsequent electrodeposition was not favorable, regardless of either increasing Fe³⁺ or Cl⁻ concentration thus being precluded (Supplementary Note 1 and Supplementary Figs. 2-6).”

Supplementary Note 1

“As shown in Supplementary Fig. 2, the Fe catalyst only demonstrated initial activity for CER in NaCl compared to NaClO₄ electrolyte, yet the CER current densities did not significantly increase with increasing either Fe³⁺ or Cl⁻ concentrations. This is because the continuous electrodeposition of Fe-catalyst film is not favorable.

In contrast to the Co-catalyst, the Fe-catalyst had two limitations. First, we noticed the dominant hydrolysis of Fe ions even at pH 2 (Supplementary Fig. 3), resulting in the decrease of soluble Fe ions. Second,

we showed that only a very small amount of Fe could be deposited on the electrode (Supplementary Fig. 4). Thus, no noticeable increase in current density was observed, and the onset potential remained almost unchanged during the 2-hour operation (Supplementary Fig. 5). Due to the limited number of deposited Fe active sites, the generated Cl₂ was little (Supplementary Fig. 6).”

The validity of the DEMS results is questionable, as the polarization curve measured during DEMS analysis showed significantly lower catalytic activity than that measured using a conventional three-electrode setup. The authors should clarify the reason for this discrepancy. Additionally, the evolution of the O₂ signal during DEMS analysis in the Cl-free electrolyte was unclear. The authors argued that O₂ evolved exclusively in these conditions, but no clear O₂ signal was observed. The reviewer assumed that the authors may have measured the signal in multiplier mode, which can induce artifacts when analytes are abundant and saturate the DEMS signal. Alternatively, the DEMS may not detect minute amounts of analyte. In this case, the multiplier mode may work properly. The absence of an O₂ signal in the Cl-free electrolyte is unusual, considering the intensity of all DEMS signals.

Response 3:

- Thanks very much for raising this critical concern.
- In this work, DEMS tests were conducted in a custom gas-tight electrochemical H-cell using a conventional three-electrode setup. In the beginning, we set a lower scan rate of 3.0 mV s⁻¹ in DEMS proposed to acquire more data points. As suggested by the referee, since this is an in situ dynamic deposition process, the scan rate will influence the deposition of Co during one potential cycle: meaning a lower scan rate leads to lower Co amount and thus low catalytic activity. As a result, the lower scan rate of 3.0 mV s⁻¹ leads to a lower amount of Co deposition and we observed lower current density in DEMS. As shown in Fig. R6, relative to 10.0 mV s⁻¹, the polarization curve measured at 3.0 mV s⁻¹ showed lower catalytic activity in the conventional three-electrode setup as well.
- To avoid misleading, we unified all the scan rates with 10.0 mV s⁻¹ and updated the DEMS data in the *main text* and *supplementary information*. Now, the results demonstrated similar current densities between DEMS and the conventional three-electrode setup (Fig. R7). The corresponding data was added in Fig. 1d on page 5 in the revised manuscript and Supplementary Fig. 24 on page S24 in the *supplementary information*.

Fig. R6 The LSV curves for CER measured in a conventional three-electrode setup in the electrolyte containing 0.1 M Co²⁺ and 0.5 M Cl⁻ under different scan rates.

Fig. R7 The LSV curves and corresponding DEMS signals during CER at pH 2 at 10 mV s^{-1} .

- For the DEMS measurements, the multiplier mode was applied (Fig. R8). During DEMS analysis, the O_2 signal in the Cl-free electrolyte was ascribed to the low current density and minute amounts of product. To make the O_2 signal clearer, we extended the test applied potential to $2.60 \text{ V}_{\text{RHE}}$ in DEMS measurements. As depicted in Fig. R9, $^{36}\text{O}_2$ was observed in the electrolyte containing 0.1 M Co^{2+} and 0.5 M ClO_4^- and no other signals were detected, such as $^{70}\text{Cl}_2$, $^1\text{H}^{16}\text{O}^{35}\text{Cl}$, $^{35}\text{Cl}^{32}\text{O}_2$, or $^{35}\text{Cl}^{16}\text{O}$. Similarly, the $^{70}\text{Cl}_2$ gas still was the only product in the electrolyte containing 0.1 M Co^{2+} and 0.5 M Cl^- during the CER process even though the test potential widens to $2.6 \text{ V}_{\text{RHE}}$ (Fig. R10). The corresponding data was supplemented in Supplementary Fig. 25 on page S25 in *supplementary information*.

Fig. R8 The instrument mode settings for DEMS experiments.

Fig. R9 The LSV curves and corresponding DEMS signals at pH 2 at 10 mV s^{-1} for OER in an electrolyte containing 0.1 M Co^{2+} and 0.5 M ClO_4^- when the test potential widens to $2.60 \text{ V}_{\text{RHE}}$.

Fig. R10 The LSV curves and corresponding DEMS signals at pH 2 at 10 mV s^{-1} for CER in an electrolyte containing 0.1 M Co^{2+} and 0.5 M Cl^- when the test potential widens to $2.60 \text{ V}_{\text{RHE}}$.

The authors did not clearly state the operating conditions for CER. Did they use a NaCl electrolyte with Co ions? If so, how did the dynamic equilibrium between Co deposition and dissolution affect CER electrocatalysis? Additionally, is this strategy industrially feasible? Alternatively, if the authors used a NaCl electrolyte free from Co ions, how did the transition metal oxide survive in such a highly corrosive environment?

Response 4:

- We are thankful to the reviewer for raising this concern. We are sorry that we did not clearly describe the operating conditions for CER leading to misleading. We quite agree with the referee. The dynamic equilibrium between Co deposition and dissolution could be established based on the applied operation condition. In our study, after the operation conditions screening, we selected an optimized condition: in an acidic electrolyte containing 0.1 M Co^{2+} and 0.5 M NaCl at pH 2. The current density will increase together with the amount of deposited Co sites and tend to saturate at the end. Once upon the variation of the applied

conditions, for instance, switching to lower potential or the open circuit potential (OCP) as shown in Fig. R11, the Co dissolution dominates thus the decrease of current density for CER. This deposition-dissolution process will initialize and refresh the catalyst.

Fig. R11 The repeatable deposition-dissolution process on the bare FTO at 1.67 V_{RHE} in the electrolyte containing 0.1 M Co²⁺ and 0.5 M Cl⁻.

- For industrial feasibility, we conducted the CER electrolysis deposited in situ in an acidic saline electrolyte containing Co²⁺ and Cl⁻ ions at 250 mA cm⁻². As shown in Fig. R12, the electrochemical activity of the CoO_xCl_y at $j_{geo}=250 \text{ mA cm}^{-2}$ only presents an overpotential of ~320 mV. This catalyst system could even outperform the representative precious and non-precious metal oxide catalysts displayed in Supplementary Table 1 on page S29 in the *supplementary information (Nat. Commun. 2020, 11, 412-422; Small 2017, 13, 1602240-1602247)*.

Fig. R12 The CER stability of CoO_xCl_y catalyst deposited in situ in an acidic saline electrolyte containing Co²⁺ and Cl⁻ ions at 250 mA cm⁻². The potential was iR corrected.

- If using a NaCl electrolyte free from Co ions, the in-situ electrodeposition and the CER cannot take place. And we quite agree with the referee and also as claimed in this manuscript, transition metal oxide cannot survive in the highly corrosive environment if we would not use this self-adaptive route.

The manuscript lacks physical characterizations after the stability test.

Response 5:

- Thanks very much for this reminder. In this revision, we have supplemented more physical characterizations after the stability test, along with more discussions to analyze the composition and structure.

The energy-dispersive X-ray spectroscopy (EDX) mappings demonstrated the presence and distribution of Co, Cl, and O for the formation of CoO_xCl_y film after the stability tests, indicating the co-deposition of Cl^- together with Co^{2+} (Fig. R13). The scanning electron microscope (SEM) images showed the dense surfaces of CoO_xCl_y films (Fig. R14).

Further, we characterized the structure of the electrodeposited CoO_xCl_y films. X-ray diffraction (XRD) displayed that the film retained an amorphous structure without characteristic diffraction peaks (Fig. R15). To gain further insight into the crystal structure of the CoO_xCl_y catalysts, high-resolution transmission electron microscopy (HRTEM) and selected area electron diffraction (SAED) were employed. The HRTEM, combined with the SAED pattern, demonstrated that the CoO_xCl_y film maintained an amorphous structure, with no discernible lattice fringes or SAED ring pattern after 500 h of the test (Fig. R16).

The supplemented data was added in Supplementary Fig. 30 on page S31, Supplementary Fig. 32 on page S33, Supplementary Fig. 35 on page S36, and Supplementary Fig. 37 on page S38 in the *supplementary information*. And the corresponding discussion has been provided on pages 7-8 in the *main text* as follows:

“The energy-dispersive X-ray spectroscopy (EDX) mappings confirmed the presence of the relatively even distribution of Co, Cl, and O for the formation of CoO_xCl_y film even after 500 h stability test (Fig. 2b and Supplementary Fig. 30).”

“The scanning electron microscope (SEM) images demonstrated the comparable dense surfaces of both CoO_xCl_y and CoO_x films (Supplementary Figs. 31-32).”

“In addition, we characterized the structure of the electrodeposited catalyst films. X-ray diffraction (XRD) demonstrated that CoO_xCl_y film was absent of characteristic diffraction peaks (Supplementary Fig. 35).”

“Even after 500 h of operation, the CoO_xCl_y film still remained the amorphous structure (Supplementary Fig. 37).”

Fig. R13 The TEM image and element mapping of CoO_xCl_y catalyst after 500 h stability test at 1.67 V_{RHE} in an electrolyte containing 0.1 M Co^{2+} and 0.5 M Cl^- .

Fig. R14 The SEM images of CoO_xCl_y catalysts after a 500 h stability test at $1.67 \text{ V}_{\text{RHE}}$ in an electrolyte containing 0.1 M Co^{2+} and 0.5 M Cl^- .

Fig. R15 The XRD images of CoO_xCl_y catalysts after a 500-h stability test at $1.67 \text{ V}_{\text{RHE}}$ in an electrolyte containing 0.1 M Co^{2+} and 0.5 M Cl^- . The FTO was selected as the background reference.

Fig. R16 The HRTEM images and corresponding SAED patterns (inset) of the CoO_xCl_y film after 500 h stability test electrodeposition at $1.67 \text{ V}_{\text{RHE}}$ in an electrolyte containing 0.1 M Co^{2+} and 0.5 M Cl^- .

Reviewer #2 (Remarks to the Author):

This work reports an amorphous CoO_xCl_y catalyst that was in situ synthesized in an acidic saline electrolyte, using Co^{2+} and Cl^- ions to adapt to the specific electrochemical conditions. The catalyst demonstrated an impressive CER selectivity of ~100% with an overpotential of approximately 0.1 V at a current density of 10 mA cm^{-2} . Notably, the catalyst remained stable for 500 hours. The addition of Cl^- to the catalyst was found to hinder the formation of highly-oxidized Co sites, which are typically required for water oxidation, while allowing for the transition from Co^* to Co-O^* as the active site for CER, ultimately leading to an enhanced CER selectivity. However, I am concerning the actual atomic structure of the active site and CER performance of the CoO_xCl_y catalyst. Although the authors employed in situ XAFS to study the valence state of Co under different CER conditions, there is a lack of evidence for the active site atomic structure. The CoO_xCl_y catalyst was prepared by in situ deposition under CER reaction potential to improve its stability. However, stability test conditions of the reported catalyst are not consistent, e.g., crystalline CoSb_2O_x stability was measured at a current density of 100 mA cm^{-2} , while $\text{RuO}_2@\text{TiO}_2$ stability was measured at a current density of 250 mA cm^{-2} , making it difficult to compare. In addition, there is still a gap between the activity of CoO_xCl_y catalyst and that of noble or non-noble metal catalysts reported in the literature, especially in terms of ultimate current density. In general, this study still needs further refinement that falls short of Nature Communications' requirements. Thus, this reviewer does not suggest the publication of this manuscript. There are some other comments on the manuscript that I hope will contribute to the successful publication of this work in the future.

Response 1:

- We highly appreciate your comments on this work, which help us improve the quality of this manuscript.
- Regarding the actual atomic structure of the active site, we have presented experimental evidence and DFT calculations in the previous manuscript. In this study, the Co catalyst consists of a single-component active site, thus the metal site has to be the Co. In situ UV-Vis spectroscopy provided experimental evidence for the coordination environment of Co metal sites and showcased the significant impact of incorporated Cl on the Co valence. The observed variations in the dominant absorption peak of Co catalyst with/without Cl indicated that the introduction of Cl could effectively tune the local structure of Co sites. And the more the Cl has been introduced, the lower the Co valence has been shown. Additionally, in situ XAFS also demonstrated the influence of Cl on the valence state of Co sites. Moreover, ex situ Raman revealed the influence of Cl on the coordination of Co sites, while XPS confirmed the presence of the Co-Cl bond. In addition, DFT calculations also indicated the impact of Cl on Co electronic charge and valence state as well, consistent with in situ UV-Vis and XAFS results. The active site transition mechanism was further proposed, which provided a comprehensive understanding of the structure of real active sites.

Besides, in response to the referee's suggestion to obtain more information about the actual atomic structure of the active site, we re-carried out more in situ XAFS experiments and analyzed the k^3 -weighted Fourier-transformed Co K-edge extended X-ray absorption fine structure (EXAFS) spectra to unfold the adjacent atomic shell around Co. As depicted in Figs. R17-18, two major peaks were observed both in CoO_x and CoO_xCl_y catalysts. The first peak at approximately $\sim 1.90 \text{ \AA}$ was attributed to Co-O coordination, while the second peak near $\sim 2.85 \text{ \AA}$ corresponded to the Co-Co coordination. Notably, the CoO_xCl_y catalyst showed an additional peak around $\sim 2.35 \text{ \AA}$, indicating the presence of Co-Cl coordination. In addition, during the anodic polarization, the CoO_xCl_y catalyst displayed a lower fitted Co-O coordination number relative to the CoO_x catalyst. Meanwhile, the CoO_xCl_y catalyst showed a larger increase in Co-O coordination number relative to the CoO_x catalyst, aligning with its larger degree of Co oxidation state change.

Fig. R17 Fitting curves of Co K-edge EXAFS in R spaces for CoO_xCl_y catalyst at (a) OCP, (b) at 1.67 V_{RHE} , (c) at 1.82 V_{RHE} , and (d) 2.02 V_{RHE} .

Fig. R18 Fitting curves of Co K-edge EXAFS in R spaces for CoO_x catalyst at (a) OCP, (b) at 1.67 V_{RHE} , (c) at 1.82 V_{RHE} , and (d) 2.02 V_{RHE} .

All EXAFS analysis has been supplemented, please find the details on page 10 in the *main text*, Supplementary Figs. 42-43 on pages S43-44, and Supplementary Tables 3-4 on pages S45-46 in the *supplementary information*. And the corresponding discussion in the *main text* as follows:

“Further, we analyzed the Fourier-transformed of k^3 -weighted Co K-edge extended X-ray absorption fine structure (EXAFS) spectra of both in CoO_x and CoO_xCl_y catalysts, which showed two major peaks associated with Co-O (~ 1.90 Å) and Co-Co (~ 2.85 Å) coordination (Supplementary Figs. 42-43). An additional peak located around ~ 2.35 Å in the CoO_xCl_y catalyst was observed, indicating the presence of Co-Cl coordination (Supplementary Fig. 43). Compared to the CoO_x catalyst, the CoO_xCl_y catalyst exhibited a lower fitted Co-O coordination number during the anodic polarization (Supplementary Tables 3-4) and displayed a larger

increase in Co-O coordination number relative to the CoO_x catalyst, in coincidence with its significant degree of Co oxidation state variation^{50,51}.”

- Regarding the stability test conditions, we performed additional CER electrolysis deposited in situ in an acidic saline electrolyte containing Co^{2+} and Cl^- ions at 250 mA cm^{-2} . This would enable a more convenient comparison of the stability at a high current. As shown in Fig. R19, the electrochemical stability of the CoO_xCl_y at $j_{\text{geo}}=250 \text{ mA cm}^{-2}$ could maintain over 20 h, which outperform the representative precious $\text{RuO}_2@\text{TiO}_2$ catalyst that showed 10 h stability measured at 250 mA cm^{-2} . The data was added in Supplementary Fig. 28 on page S28 and Supplementary Table 1 on page S29 in the *supplementary information* and the corresponding discussion has been provided on page 7 in the *main text* as follows:

“Meanwhile, the electrochemical stability of the CoO_xCl_y at $j_{\text{geo}}=250 \text{ mA cm}^{-2}$ could sustain over 20 h (Supplementary Fig. 28).”

Fig. R19 (a) long-term stability test of three $\text{RuO}_2@\text{TiO}_2$ electrodes under the same geometric current density (250 mA cm^{-2}) (*Small* 2017, 13, 1602240-1602247). (b) The CER stability of CoO_xCl_y catalyst deposited in situ in an acidic saline electrolyte containing Co^{2+} and Cl^- ions at 250 mA cm^{-2} . The potential was iR corrected.

- Regarding the activity comparison, firstly, we assessed the intrinsic potential of different catalysts, including those reported in the literature. These potentials at 1.0 mA cm^{-2} (the electrochemically active surface area was used) were determined from CV data. For our CoO_xCl_y catalyst, the electrochemically active surface area was calculated by the double-layer capacitance measurements (Fig. R20), and the intrinsic potential at 1.0 mA cm^{-2} was subsequently obtained. As illustrated in Fig. R21, the intrinsic potential at 1.0 mA cm^{-2} of the CoO_xCl_y catalyst is comparable to the noble metal catalyst and lower than the non-noble metal catalysts.

Fig. R20 Double-layer capacitance measurements. **(a)** CV curves in 0.5 M Cl^- electrolyte in a non-Faradic region (0.965~1.065 V_{RHE}) with a scan rate of 10 mV s^{-1} to 90 mV s^{-1} . **(b)** Liner fitting of the cathodic and anodic charging current density at 1.015 V_{RHE} as a function of scan rate. The CoO_xCl_y catalyst on FTO was obtained after 2.0 h deposited in situ in an acidic electrolyte containing 0.1 M Co^{2+} and 0.5 M Cl^- ions.

Fig. R21 The intrinsic potential of different catalysts determined from CV data at 1.0 mA cm^{-2} of electrochemically active surface area. The CoO_xCl_y catalyst is in our work. The NiSb_2O_x , CoSb_2O_x , MnSb_2O_x , RuTiO_x , and Co_3O_4 catalysts are reported in the literature (*Energy Environ. Sci.*, 2019, 12, 1241-1248; *J. Mater. Chem. A*, 2018, 6, 12718-12723).

Secondly, for the ultimate current density, even though the CoO_xCl_y catalyst is a unary metal catalyst with a single-component active metal site, the CoO_xCl_y catalyst exhibited a superior activity of 250 mA cm^{-2} only at 1.68 V_{RHE} in 0.5 M NaCl (Fig. R19). However, the noble metal catalyst, such as nanostructured $\text{RuO}_2\text{-TiO}_2$ electrodes exhibit ~2.16 V_{RHE} at 250 mA cm^{-2} (*Small* 2017, 13, 1602240-1602247). In addition, the multi-metal NiSb_2O_x , CoSb_2O_x , MnSb_2O_x , and noble-metal RuTiO_x in 4.0 M NaCl (*Energy Environ. Sci.*, 2019, 12, 1241-1248) exhibited the current density of 100 mA cm^{-2} at ~1.96 V_{RHE} , ~1.85 V_{RHE} , ~2.1 V_{RHE} , ~1.8 V_{RHE} , as depicted in Fig. R20. We also found that the ultimate mass activity of the CoO_xCl_y catalyst after 2 h electrodeposition could arrive the 360 $\text{mA cm}^{-2} \text{ug}^{-1}$. These comparisons showcased the favorable performance of the CoO_xCl_y catalyst, positioning it favorably among the catalysts studied.

In general, the CoO_xCl_y catalyst as non-noble metal oxides could exhibit superior performance with ~100% selectivity for sustainable chlorine evolution in acid brine. Additionally, this study provides fundamental insights into how the reactant Cl itself can work as a promoter toward enhancing CER in acidic brine, which could inspire the use of seawater as an electrolyte for electrocatalysis and promote the development of this field. Thus, this work should be urgent to fill that knowledge gap.

Fig. R22 (a) Initial electrochemical behavior of NiSb₂O_x, CoSb₂O_x, MnSb₂O_x, and RuTiO_x in pH = 2.0, 4.0 M NaCl(aq) electrolyte (*Energy Environ. Sci.* 2019, 12, 1241-1248). (b) The LSV of CoO_xCl_y catalyst after 500 h stability tests in 0.1 M Co²⁺ + 0.5 M Cl⁻ at pH = 2. The potential was iR corrected.

Fig. R23 The mass activity of CoO_xCl_y catalyst after 2 h deposition.

1. It is recommended to incorporate the LSV test following catalyst activation. Furthermore, a comparison of the activity, selectivity, and stability of the catalyst with RuO₂ and DSA is advisable.

Response 2:

- We sincerely thank the reviewers for their suggestions. We have supplemented the LSV data of the CoO_xCl_y catalyst after 500 h electrodeposition at 1.67 V_{RHE} in the electrolyte containing 0.1 M Co²⁺ and 0.5 M Cl⁻ (Fig. R24). The data was added in Supplementary Fig. 27 on page S27 and the corresponding discussion has been provided on page 7 in the *main text* as follows:

“In contrast, the current density of the CoO_xCl_y catalyst deposited at 1.67 V increases to 10 mA cm⁻² and then stabilizes at about 15 mA cm⁻² over 500 h test, which leads to a downshift of potential ~ 300 mV at 10 mA cm⁻² relative to the initial CV activity (Supplementary Fig. 27).”

Fig. R24 The CV curves of the CoO_xCl_y film after 500 h electrodeposition 1.67 V_{RHE} in an electrolyte containing 0.1 M Co^{2+} and 0.5 M Cl^- .

- Further, we conducted evaluations of the activity, selectivity, and stability of both RuO_2 and DSA catalysts. It is worth mentioning that the DSA catalyst was typically loaded on Ti, which itself demonstrated activity for the CER as reported in *Nature* (*Nature*, 2023, 617, 519-523). Due to this consideration, we chose to employ FTO glass as the support material for the intrinsic performance evaluation, consistent with the measurements in our work. The commercial RuO_2 and DSA (Ir/Ru=2) were loaded on FTO and their performance was assessed.

Compared with the CoO_xCl_y catalyst, the DSA and RuO_2 catalysts did not obviously shift the onset potential in 0.5 M NaCl electrolyte relative to the Cl^- -free electrolyte (Fig. R25). And the overpotential at 10 mA cm^{-2} was ~350 mV and ~370 mV, respectively, which were significantly higher than the CoO_xCl_y catalyst.

Fig. R25 The CV curves of different catalysts for both CER and OER. (a) RuO_2 , (b) DSA. The electrolysis for CER operated in 0.5 M NaCl electrolytes at $\text{pH} = 2$. The electrolysis for OER operated in 0.5 M NaClO_4 electrolytes at $\text{pH} = 2$.

In addition, the current density of the DSA catalyst quickly declined from ~8.5 mA cm^{-2} to ~0.6 mA cm^{-2} with ~93% loss in activity during the 50 h stability test at 1.67 V_{RHE} (Fig. R26). And RuO_2 catalyst experienced a decrease in activity of ~89% during 100 h stability tests. These results indicated that both the DSA and RuO_2 catalysts demonstrated a substantial loss in their catalytic activities for the stability tests.

Fig. R26 The CER stability for different catalysts. (a) CV curves of the RuO₂ electrode before and after 50 h potentiostatic operation at 1.67 V_{RHE} in 0.5 M NaCl electrolyte and (b) 50 h of potentiostatic operation. (c) CV curves of the DSA electrode before and after 100 h potentiostatic operation at 1.67 V_{RHE} in 0.5 M NaCl electrolyte and (d) 100 h of potentiostatic operation.

Further, we evaluated the CER selectivity of DSA and RuO₂ catalysts using RRDE in an Ar-saturated electrolyte with different concentrations of Cl⁻ at pH 2. In Fig. R27, it was evident that the selectivity of DSA and RuO₂ catalysts was distinctly lower than that of CoO_xCl_y even in 0.5 M NaCl electrolyte. Besides, their selectivity decreased as the applied potentials increased.

Fig. R27 The CER selectivity for different materials. The Cl₂ selectivity of different catalysts under different applied anodic potentials and Cl⁻ concentrations of (a) 0.1 M and (b) 0.5 M during the CER process at pH 2. The data were recorded based on a rotating disk electrode method at 1600 RPM.

The corresponding discussion has been provided on pages 4, 6, and 7 in the *main text* and the data was added in Supplementary Fig. 7 on page S5, Supplementary Fig. 22 on page S22, and Supplementary Fig. 26 on page S26 in the *supplementary information*. The catalyst preparation method was updated in the experimental

section on page 17 in the *main text* as follows:

“**Synthesis of RuO₂ and DSA catalysts.** To prepare the RuO₂ film electrode, 5 mg commercial RuO₂ was dissolved into 0.95 mL ethanol by adding 0.05 mL 5 wt% nafion117 solution under ultrasonic treatment for 60 min. 10 uL of this suspension was dropped onto the clean FTO (the efficient contact area is 0.283 cm²) for the formation of a catalyst film. The DSA catalyst film was prepared based on the commercial protocol on Ti (the ratio of Ir/Ru is 2).”

2. The Co K α and Cl K α mapping of CoO_xCl_y catalyst electrodes by XRF shows that Co and Cl elements are distributed throughout the catalyst film, but their concentration varies across the film. Figure 2a shows that Co is more concentrated in the middle, while Cl is concentrated at the edge. Is this reasonable?

Response 3:

- Thanks very much for raising this question. We apologize for this mistake. In the previous manuscript, the color mapping with the multi-color mode was initially intended to showcase the element distribution. However, we inadvertently shortened the numerical range of the color bar, which resulted in the filtering of most data points, leading to significant visual color differences. To address this issue, we re-processed the Co K α and Cl K α mapping using the single-color mode based on the previous data. As depicted in Fig. R28, we observed that the Co and Cl elements were distributed throughout the entire catalyst film. It's important to note that the background signal in the Cl K α mapping might be attributed to the low amount of Cl element, along with the lower X-ray detection and collection efficiency for Cl due to its light element with a low atomic number. Additionally, we repeated XRF experiments as shown in Fig. R29. The Co K α and Cl K α mapping of the CoO_xCl_y catalyst confirmed that the Co and Cl elements were evenly distributed throughout the catalyst film. The updated corresponding data can be found in Fig. 2a on page 8 of the revised manuscript.

Fig. R28 The XRF Co K α and Cl K α mapping of CoO_xCl_y catalyst. The CoO_xCl_y film was electrodeposited in the electrolyte containing 0.1 M Co²⁺ and 0.5 M Cl⁻ at 1.67 V_{RHE} for 10 h.

Fig. R29 The XRF Co K α and Cl K α mapping of CoO_xCl_y catalyst. The CoO_xCl_y film was electrodeposited in the electrolyte containing 0.1 M Co²⁺ and 0.5 M Cl⁻ at 1.67 V_{RHE} for 10 h.

3. The authors used XRF and XPS to characterize the changes in Cl content as "Analogous to Co (Supplementary Fig. 8 and 26a), the content of Cl increased with the electrodeposition time (Supplementary Fig. 26b)". Can the authors give the magnitude or range of x and y in CoO_xCl_y ? This is informative for the reasonableness of the theoretical calculation modeling.

Response 4:

- Thanks very much for this suggestion. We have provided the additional XPS data on the Co, Cl, and O contents at different deposition times (Table. R1). The results showed that the ratio range of Co/Cl was about 10/1~5/1, and the Co/O ratio was approximately 1/4~1/3. The range of x and y in CoO_xCl_y was selected for the theoretical calculation modeling, validating the accuracy of our DFT calculations. The corresponding data was added in Supplementary Table 2 on page S40 in the *supplementary information*.

Table. R1 The contents of Co, Cl, and O at different deposition times by XPS test.

Time/h	Co/at %	Cl/at %	O/at %	CoO_xCl_y
0.5	11.67	1.13	43.67	$\text{CoO}_{3.74}\text{Cl}_{0.1}$
1.0	12.60	1.31	41.33	$\text{CoO}_{3.7}\text{Cl}_{0.1}$
2.0	13.91	3.26	42.21	$\text{CoO}_{3.0}\text{Cl}_{0.23}$
5.0	15.60	6.83	43.32	$\text{CoO}_{2.74}\text{Cl}_{0.44}$

4. The amorphous structure of CoO_xCl_y film in the paper is only demonstrated by HRTEM images and corresponding SAED patterns. Can other characterization techniques, such as X-ray diffraction (XRD), be used in conjunction with other methods to increase the level of confidence in the amorphous structure of CoO_xCl_y catalysts, particularly after activation?

Response 5:

- Thanks for this suggestion. We have incorporated X-ray diffraction (XRD) analysis to further enhance our understanding of the CoO_xCl_y films. The data revealed that the CoO_xCl_y film did not exhibit any characteristic diffraction peaks (Fig. R30), indicating its amorphous nature. Additionally, after conducting the HRTEM, SAED patterns, and XRD characterizations on the CoO_xCl_y catalyst following the 500-hour stability tests, we observed that the film retained its amorphous structure (Fig. R31). These results align with our expectations, demonstrating the film's amorphous structure even after prolonged deposition.

The data was added in Supplementary Fig. 35 on page S36 and Supplementary Fig. 37 on page S38 in the *supplementary information* and the corresponding discussion has been provided on pages 7-8 in the *main text* as follows:

"In addition, we characterized the structure of the electrodeposited catalyst films. X-ray diffraction (XRD) demonstrated that CoO_xCl_y film was absent of characteristic diffraction peaks (Supplementary Fig. 35)."

"Even after 500 h of operation, the CoO_xCl_y film still remained the amorphous structure (Supplementary Fig. 37)."

Fig. R30 The XRD images of CoO_xCl_y catalyst after 40 h deposition at $1.67 V_{\text{RHE}}$ in an electrolyte containing 0.1 M Co^{2+} and 0.5 M Cl^- . The FTO was selected as the background reference.

Fig. R31 The structure of CoO_xCl_y catalyst after 500 h deposition. **(a)** XRD images. The FTO was selected as the background reference. **(b)** The HRTEM images and corresponding SAED patterns (inset) of the CoO_xCl_y film. The 500 h stability deposition was carried out at $1.67 V_{\text{RHE}}$ in an electrolyte containing 0.1 M Co^{2+} and 0.5 M Cl^- .

5. The region chosen in Figure 3d may be not reasonable, and it is suggested to reconfirm the absorption energy (E_0), obtained from the first maximum in the first order derivative as the electronic vacancy, before determining the oxidation state of Co. It is suggested to refer to *Angew. Chem. Int. Ed.* 2022, e202209486 and *J. Am. Chem. Soc.* 2017, 139, 12076-12083. In addition, as seen from the intensities of the white-line peaks in Figure 3d, the CoO_xCl_y valence at 1.67 V is the lowest, which is not consistent with the authors' conclusion.

Response 6:

- Thanks very much for the suggestions. We have carefully studied the suggested work on XAFS (*Angew. Chem. Int. Ed.* 2022, e202209486 and *J. Am. Chem. Soc.* 2017, 139, 12076-12083). The absorption energy was updated and referred to the first maximum in the first order derivative in XANES data (Figs. R34-35, vide infra), and the detailed methods were supplemented to the experimental section.

We thank the referee for making us aware of some recent works on XAFS. We have now considered these very valuable publications and referred to them where are suitable.

We have added (references 47 and 59) and referred to them on page 10 and page 18 in the *main text*:

“The Co K-edge X-ray absorption near edge structure spectra (XANES) of CoO_xCl_y and CoO_x film were collected under anodic polarization in the 0.5 M NaCl and NaClO_4 electrolytes, respectively⁴⁷.”

“The E_0 was obtained from the largest peak in the 1st derivative XANES according to the previous work^{47,59}.”

- Regarding the intensities of the white-line peaks observed in the CoO_xCl_y catalyst, it is important to consider the potential influence of background processing during the normalization of XAFS data. As shown in Fig. R32, the pre-edge line of the CoO_xCl_y catalyst at 1.67 V_{RHE} exhibited a significant deviation from the others, indicating a potential impact on the intensity of the white-line peak. To ensure the credibility of our data, we re-conducted in situ XAFS spectroscopy (Fig. R33-35). Through this updated Co K-edge XANES analysis, we found that the intensities of the white-line peaks remained consistent with the variation in anodic potential. Consequently, we have revised and consolidated the relevant data in Fig. 3d on page 9 of the *main text* and Supplementary Figs. 40-41 on pages S41-42 in the *supplementary information*.

Fig. R32 The enlarged Co K-edge XANES data without normalization of CoO_xCl_y catalyst after different potential polarization.

Fig. R33 The images of (a) the in situ XAFS equipment and (b) the electrochemical cell.

Fig. R34 Co K-edge XANES at different applied potentials from the open-circuit condition to 2.02 V_{RHE} for CoO_xCl_y film.

Fig. R35 Co K-edge energy at different applied potentials for CoO_x and CoO_xCl_y film. The absorption energy (E_0) is obtained from the first maximum in the first-order derivative XANES.

6. The equilibrium potential of CER is correlated with temperature, chloride ion activity and pH, and the authors need to reconfirm the equilibrium potential in this paper.

Response 7:

- Thanks very much for this kind comment. We agree with the referee that the equilibrium potential of CER is correlated with temperature, chloride ion activity, and pH. Our previous description “*Even though the equilibrium potential of 1.36 V for CER is 0.13 V higher than the equilibrium potential of 1.23 V for OER*” was indeed not strict on page 14 in the *main text*. Here, the potential of 1.36 V for CER is the standard theoretical electrode potential obtained from the experiment (vs SHE). On the potential scale of reversible hydrogen electrodes (RHE), CER’s equilibrium potential is correlated with temperature, chloride ion activity, and pH. In our work, we took into account the effect of chloride ion activity and pH on the potential scale of RHE. Hence, we corrected the description of the equilibrium potential of CER to the theoretical potential of CER on page 14 in the *main text*.

Reviewer #3 (Remarks to the Author):

This paper by Xiao et al. addresses the very challenging issue that non-precious catalysts cannot stand for selective chlorine evolution reaction in the very corrosive and strong acid. The authors proposed an idea that allows the catalyst deposition to adapt to the hostile electrochemical conditions, with an unexpected amorphous CoO_xCl_y catalyst in situ deposited on electrodes in acidic brine containing Co^{2+} ions. This resultant catalyst presents 100% selectivity and long-term stability over 500 hours. In combination in-situ spectroscopies with theoretical calculations, the authors find the key role of reactant Cl^- itself as a promoter for catalyst deposition and chlorine evolution reaction. Overall, the authors have done a nice work since it can be a reference for acidic sea water splitting that produces pure chlorine gas rather than oxygen on the anodes. Before I could recommend it for publication in Nature Communications, the authors should address a few issues that I have highlighted below.

Responses 1:

- Thanks very much for your positive comments and constructive suggestions on our work. We responded to all the questions one by one as follows and revised the manuscript accordingly.

1. page 5 Fig. 1b – After a certain time of electrodeposition process, the catalyst seems to approach to the lowest overpotential. The authors may specify which part is the adaptive deposition process and suggest the possibility to accelerate this process?

Responses 2:

- Thanks very much for raising this question.
- In this study, the Cl^- not only aids the in situ deposition of the CoO_xCl_y catalyst but also suppresses the formation of a higher Co oxidation state thus enhancing activity and selectivity for chloride oxidation. Here, the adaptive deposition process refers to the amorphous CoO_xCl_y catalyst that has been deposited in situ in an acidic saline electrolyte containing Co^{2+} and Cl^- ions under a given electrochemical condition. During in situ deposition process, the catalyst film presents an epitaxial-growth-like behavior, of which the applied potential at the electrode/electrolyte interface can oxidize the Co^{2+} for the deposition of the outermost catalyst layer and the interfacial potential drop across the catalyst with the gradual introduction of Cl^- ions to adapt the electrocatalytic process and the final state of the deposited film depends on a given electrochemical condition, including the applied potential, solution pH, electrolyte component, and concentration. The CER sites were expected to self-optimize to the best stable state.
- Concerning the possible methods to accelerate this process, we increase the deposition potential and found that the performance after short-time high potential deposition is similar to the long-time deposition at low potential. The catalyst deposition rate depends on the applied potentials. A fast catalyst deposition can be obtained at a higher applied potential. As shown in Fig. R36, when the deposition potential increases at 1.82 V_{RHE} , the current density of the CoO_xCl_y catalyst deposited increases to 15 mA cm^{-2} and then stabilizes at about 20 mA cm^{-2} after the 5.0 h test. And then, the current density could arrive at about 10 mA cm^{-2} when the deposition potential return to 1.67 V_{RHE} .

Fig. R36 The CER activity at different applied potentials. The electrolyte at pH = 2 contains 0.1 M Co^{2+} and 0.5 M Cl^- ions.

2. page 5 Fig. 1c – The authors applied the RRDE to measure the Cl_2 selectivity at different potentials in acidic electrolyte containing Co^{2+} . A potential of 0.95 V was applied on Pt ring. How to exclude the possible deposition of Co on Pt ring at this potential?

Responses 3:

- Thanks very much for raising this concern. We supplemented the Inductive Coupled Plasma Mass Spectrometer (ICP-MS) analysis to detect the possible presence of Co on the Pt ring. The results from the ICP-MS analysis indicated that there was no Co deposition on the Pt ring when using the chronoamperometry (CA) and linear sweep voltammetry (LSV) methods at 0.95 V, as compared to the initial electrode condition (Fig. R37).

Fig. R37 The content of Co on Pt ring after different test methods.

3. page 6, lines 142 - 143 and page S21, Supplementary Fig. 19 – the initial current density $\sim 5.0 \text{ mA cm}^{-2}$ of CoO_xCl_y at 1.67 V before the deposition process is closed to Co_3O_4 . It seems that there is a very rapid deposition of CoO_xCl_y . What are the mass loadings of Co_3O_4 and CoO_xCl_y catalysts?

Responses 4:

- Thanks very much for this question. We conducted the ICP-MS characterization to assess the mass loading of Co_3O_4 and CoO_xCl_y catalysts. As shown in Fig. R38, the Co content of the Co_3O_4 catalyst is about 180 times higher than that of the CoO_xCl_y catalyst at the same current density.

Fig. R38 The mass loading of Co₃O₄ and CoO_xCl_y catalysts. **(a)** The activity of 30 min potentiostatic operation at 1.67 V in 0.1 M Co²⁺ + 0.5 M Cl⁻ electrolyte. **(b)** The mass loading of Co₃O₄ and CoO_xCl_y catalysts by ICP-MS.

4. page 7, line 161 – there is a typo - “Supplementary Figs. 23” should be “Supplementary Fig. 23”.

Responses 5:

- Thanks very much for the reminder. We corrected it on page 6 in the *main text*.

5. page 8, Fig. 2a – the color bar is missing.

Responses 6:

- Thanks very much for pointing out this question. We remeasured the data and updated it in Fig. 2a on page 8 of the revised manuscript.

6. page 13, line 278; page 14, line 280 – better using “theoretical overpotential” in DFT calculations.

Responses 7:

- Thanks very much for reminding us. We corrected the description of the theoretical overpotential for DFT calculations on pages 14-15 in the *main text*.

7. page S1, lines 30-31 – “The electrolyte at pH = 2 containing 0.001 M indicated transition metal salts and 0.5 M NaCl (for CER)/NaClO₄ (for OER)”. The description is confusing.

Responses 8:

- Thanks very much for raising this concern. The corresponding description was corrected on page S1 in the *supplementary information* as follows:

“The CER electrolysis was operated in 0.5 M NaCl electrolytes at pH 2 containing 0.001 M metal nitrates. The OER electrolysis was operated in 0.5 M NaClO₄ electrolytes at pH 2 containing 0.001 M metal nitrates.”

8. page S12-S17, Supplementary Fig. 12-15 – the error bar is confusing. The calculated Cl₂ selectivity at the corresponding current density/potential should be constant, the error bars obtained by repeated measurements have already been shown in Fig. 1c.

Responses 9:

- Thanks very much for raising this concern. We double checked the error bar and corrected them on pages

S18-21 in the revised *supplementary information*.

9. page S23, Supplementary Fig. 20 – the scale bar is missing.

Responses 10:

- Thanks very much for the reminder. We supplemented the scale bar in Supplementary Fig. 29 on page S30 in the revised *supplementary information*.

10. page S23, Supplementary Fig. 20b – the authors may provide the contrasting background of Cl K α mapping.

Responses 11:

- Thanks very much for your suggestion. The contrasting background picture was added in Supplementary Fig. 29b on page S30 in the *supplementary information*.

11. page S32, Supplementary Table 2 – there are two typos. “Co³⁺ and Co⁴⁺” should be “Co²⁺ and Co³⁺”.

Responses 12:

- Thanks very much for pointing out this question. We recheck carefully and corrected it in Supplementary Table 5 on page S48 in the revised *supplementary information*.

REVIEWERS' COMMENTS

Reviewer #1 (Remarks to the Author):

This reviewer's questions and comments have been properly addressed. There are no more questions from this reviewer.

Reviewer #2 (Remarks to the Author):

The revised manuscript has adequately address all key questions raised by me. I therefore would like to recommend for acceptance of the manuscript in its current form for publication.

Reviewer #3 (Remarks to the Author):

All my concerns have been well addressed and I suggest the publication of this manuscript.